# ENHANCING ZEROTH-ORDER FINE-TUNING FOR LLMS VIA GRADIENT-GUIDED SUBSPACE SELECTION

## ABSTRACT

As a promising memory-efficient technique, zeroth-order (ZO) optimization enables large language models (LLMs) to bypass the costly process of backpropagation during fine-tuning by estimating gradients through function evaluations. However, to minimize approximate variance in high-dimensional parameter spaces, existing ZO methods focus on exploring the estimate of gradients within random subspaces, neglecting the benefits of searching for more accurate subspaces of LLMs on gradient estimates. Due to inaccurate gradient estimates obtained from random spaces, fine-tuning performance is inevitably degraded, thus compromising the performance of downstream tasks. To address the limitation of existing ZO methods, this paper proposes a novel ZO subspace fine-tuning method named *SVD-0*. Based on singular value decomposition (SVD), SVD-0 can effectively obtain more accurate subspace projection matrices, which can be used to improve the accuracy of gradient estimates. Experimental results on various language modeling tasks show that SVD-0 achieves better fine-tuning performance than SOTA ZO methods.

## 1 INTRODUCTION

Due to the powerful capabilities of language understanding and reasoning, large language models (LLMs) have demonstrated significant performance on a wide range of tasks, such as mathematical reasoning (Guo et al., 2025), creative writing (Shanahan & Clarke, 2023). Currently, fine-tuning (FT) the pre-trained foundation model to adapt to downstream tasks has become the mainstream paradigm for AI application development. However, due to the extremely large number of model parameters, traditional first-order (FO) optimization-based fine-tuning methods face a serious challenge of excessive memory consumption. Typically, since the backpropagation process in FO requires storing activations and optimizer states, the memory requirements of FT are significantly larger than those of reasoning, which severely limits the development of LLM-based applications.

To achieve memory-efficient FT, existing methods can be classified into two categories, i.e., parameter-efficient fine-tuning (PEFT) methods (Liu et al., 2022; Han et al., 2024) and zeroth-order (ZO) optimization methods (Malladi et al., 2023). PEFT methods attempt to reduce the number of trainable parameters to alleviate memory requirements. However, since PEFT methods are still based on FO optimization, they require a significant amount of memory to store intermediate training results, which severely limits the choice of trainable parameters. ZO optimization methods (Malladi et al., 2023) emerge as a promising alternative by estimating gradients through forward-pass perturbations, thereby eliminating the memory overhead associated with backpropagation. However, conventional ZO methods face a critical challenge: the high variance of gradient approximations in billion-parameter spaces severely degrades optimization efficiency and model performance.

Recent advances in ZO optimization for LLMs, such as SubZero (Yu et al., 2024) and LOZO (Chen et al., 2025), attempt to mitigate this issue by constraining perturbations to random low-dimensional subspaces. These methods are based on the finding that gradient matrices become low-rank during LLM training and fine-tuning (Zhao et al., 2024a). While these subspace methods reduce approximation variance, they fundamentally rely on arbitrary projection matrices that fail to match the low-rank structure implied by the gradient. This limitation stems from a fundamental disconnect - the subspace construction process ignores critical gradient information that could guide more effective parameter updates. Therefore, *how to determine the optimal low-dimensional subspaces without relying on first-order optimizers* poses a fundamental challenge.

The similarity between the gradient estimated by ZO optimizers and the true gradient has been experimentally demonstrated in (Malladi et al., 2023). We find it feasible to derive the low-rank structure of the true gradient from the estimated gradient. In light of this idea, we conducted a preliminary study (see details in Section 3). The experimental results indicate a significant similarity between the estimated and true gradients, as demonstrated by the resemblance of their singular value vectors. Consequently, we conclude that applying singular value decomposition (SVD) to the gradient estimated by the ZO optimizer allows us to obtain a low-rank structure that closely resembles the low-rank structure of the true gradient.

Motivated by the above findings, we propose SVD-0, a novel gradient-guided subspace optimization framework that combines zeroth-order efficiency with principled subspace discovery. Our key insight is that, while exact first-order gradients remain inaccessible due to memory constraints, ZO gradient estimates contain sufficient directional information to reconstruct high-fidelity subspaces. Specifically, SVD-0 periodically performs singular value decomposition (SVD) on ZO gradient estimates to derive layer-wise projection matrices that capture dominant optimization directions. By preserving the intrinsic structure of the subspace, our method effectively enhances the performance of subspace-based ZO methods. The contributions of this work are summarized as follows:

- We propose a novel method for exploring more accurate subspace projection matrices and conducting layer-wise perturbations on low-rank matrices. With periodic updates of the projection matrices, our method continuously captures the subspaces of the parameters.
- We develop a novel gradient-guided ZO method to approximate these two projection matrices, ensuring low memory usage throughout the entire fine-tuning process, to overcome the paradox that obtaining subspace projection matrices requires FO gradients.
- We conduct comprehensive experiments on various model scales and language modeling tasks. The corresponding results demonstrate the superiority of our method over various ZO optimization methods specifically designed for LLM fine-tuning.

## 2 RELATED WORK

**Memory-efficient fine-tuning for LLMs.** Recent work has concentrated on exploring memory-efficient fine-tuning methods to enable LLM fine-tuning on memory-intensive hardware. A critical line of research centers on Parameter-Efficient Fine-Tuning (PEFT) methods (Liu et al., 2022; Han et al., 2024) by freezing the backbone of LLMs while only tuning a small group of parameters. For instance, LoRA (Hu et al., 2022) only updates parameters based on low-rank structures while being competitive with full-parameter fine-tuning. LISA (Pan et al., 2024) distinguishes trainable layers based on their contribution to task-specific performance and freezes other layers to reduce the memory footprint. Further, parameter quantization (Lin et al., 2024; Frantar et al., 2022) has played a pivotal role in enhancing memory efficiency. By discretizing model parameters (e.g., from 32-bit to 8-bit or lower precision), quantization methods such as QLoRA (Dettmers et al., 2023) and LLM.int8() (Dettmers et al., 2022) reduce storage requirements without significant degradation in task performance. Complementary to PEFT and the quantization method, subspace projection techniques have emerged as a powerful strategy to reduce the dimensionality of the optimization space. Galora (Zhao et al., 2024a) and FLORA (Hao et al., 2024) both leverage the low-rank property of gradients to constrain updates on a compact subspace of the full parameter space (Huang et al., 2025). By discovering the projection matrices of low-rank subspaces, the memory costs for storing gradients and optimizer states (e.g., the first and second order states in Adam optimizer (Kingma & Ba, 2014)) are greatly reduced.

**Zero-order optimization.** ZO approaches enable backpropagation-free optimization by approximating exact gradients through finite differences. This flexibility has driven interest in ZO for solving a range of machine learning problems, including on-chip learning, black-box adversarial strategies, and memory-efficient LLMs fine-tuning (Malladi et al., 2023; Zhang et al., 2024). Despite these strengths, the practical application of ZO is primarily limited to smaller-scale tasks and models. A critical limitation stems from the high error in its gradient approximations (Park et al., 2025), which becomes more pronounced as problems grow larger and more complex, making scaling particularly challenging. To address this issue, approaches such as MeZO-SVRG (Gautam et al., 2024) and DiZO (Tan et al., 2025) utilize variance-reduction methodologies (Ma & Huang, 2025) to mitigate gradient divergence. Furthermore, methods including SparseMezo (Liu et al., 2024), TeZO (Sun et al.,

2025), and AdaZeta (Yang et al., 2024) have been proposed to diminish approximation errors by reducing dependence on the parameter dimension through parameter sparsification and tensorization. Subspace methods (Nozawa et al., 2025), including SubZero (Yu et al., 2024) and LOZO (Chen et al., 2025), are explored to leverage low-rank structures for decreasing the error. Although they effectively alleviate the variance of gradient approximation, the randomly generated projection matrices cannot precisely reflect the transformation between the subspace and the full space, leading to degradation in model performance.

## 3 PRESTUDY

In exploring the alignment between estimated ZO and true FO gradients in the parameter spaces of large language models, we perform a targeted analysis using the OPT-1.3B model (Zhang et al., 2022) on the RTE task (Dagan et al., 2005; Bar Haim et al., 2006; Giampiccolo et al., 2007; Bentivogli et al., 2009). For every 50 training steps, we determine the exact FO gradients through backpropagation with a batch size of 16 and ZO gradient estimates via MeZO's simultaneous perturbation method (Malladi et al., 2023). Subsequently, we apply singular value decomposition (SVD) to both gradient matrices. We then assess the cosine similarity between the singular value vectors.

Figure 1 illustrates that the singular vectors demonstrate high cosine similarity. This finding indicates that the ZO gradients maintain critical optimization directions and exhibit a similar low-rank structure. This supports our main hypothesis that ZO gradient estimates contain sufficient spectral information to reconstruct low-rank subspaces guided by FO methods. The preserved accuracy in directional estimates suggests that by limiting ZO perturbations to the primary gradient subspaces, we can reduce approximation variance while still achieving effective updates. These concepts form the foundation of our SVD-0 optimization framework, which systematically leverages the

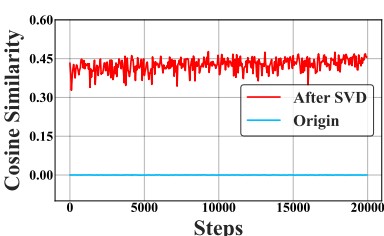

Figure 1: Cosine similarities between estimated ZO gradients and true gradients for "Origin" and "After SVD".

inherent structure in ZO gradient estimates to achieve FO-guided efficiency without the computational overhead associated with backpropagation. Additional experiments can be found in Appendix B.

## 4 METHODOLOGY

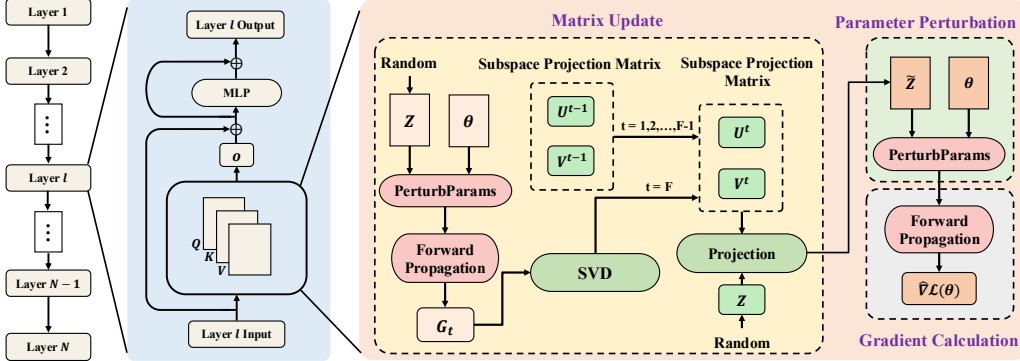

Figure 2: Framework and workflow of our SVD-0 method.

Figure 2 illustrates our approach, which focuses on two main components: the matrix update module and the parameter perturbation module. The matrix update module is for computing and adjusting the projection matrices, represented as $U \in \mathbb{R}^{m \times r}$ and $V \in \mathbb{R}^{n \times r}$. Together with a low-dimensional random matrix $Z \in \mathbb{R}^{r \times r}$, these matrices are used to generate a low-rank perturbation $\tilde{Z}$.

Within the first module (i.e., the matrix update module), we introduce an innovative and precise approach to acquire the matrices $U$ and $V$, as detailed in Algorithm 1. Traditional approaches often utilize random low-rank perturbation matrices (Chen et al., 2025; Yu et al., 2024). This randomness contributed to uncertainty in the gradient update process during training. In contrast,

our approach computes the $U$ and $V$ matrices based on the gradient information derived using the MeZO method (Malladi et al., 2023) before each update.

The second module serves to perturb the parameters, as described in Algorithm 2. Common enhancements, such as SubZero (Yu et al., 2024) and the SVD-0 approach proposed here, reformulate the update mechanism by adopting a low-rank perturbation method. As illustrated in Figure 2, the low-rank perturbation $\tilde{Z} \in \mathbb{R}^{m \times n}$ is determined in the following manner:

$$\tilde{Z} = UZV^T, \tag{1}$$

where $Z \in \mathbb{R}^{r \times r}$ is a random perturbation matrix sampled from $\mathcal{N}(0, 1)$. Consequently, the parameter $\theta_t \in \mathbb{R}^{m \times n}$ during the $t^{th}$ iteration is determined by $\theta_t^{\pm} = \theta \pm \tilde{Z} = \theta \pm UZV^T$. Thus, the gradient is approximated using two forward evaluations as expressed below:

$$\widehat{\nabla}\mathcal{L}(\theta_t^{\pm}) = \frac{\mathcal{L}(\theta_t^+; \mathcal{B}) - \mathcal{L}(\theta_t^-; \mathcal{B})}{2\epsilon} UZV^T. \tag{2}$$

### 4.1 GRADIENT-GUIDED SUBSPACE PROJECTION MATRIX ACQUISITION

Existing approaches to projection matrix construction consist of a spectrum of techniques, ranging from randomized sampling methods (Chen et al., 2025; Yu et al., 2024) to computationally intensive deterministic algorithms (Zhao et al., 2024b). Although the former is computationally efficient, it has the drawback of insufficient approximation accuracy due to its reliance on randomness. The latter introduces significant computational overhead while not significantly improving the approximation accuracy. To address this limitation, we propose a balancing strategy based on adaptive subspace decomposition, as shown in lines 4-7 of Algorithm 3.

---

**Algorithm 1** GenerateProjMatrix($G, r$)

**Input**: i) $G$, estimated gradient of parameter matrix; ii) $r$, rank.
**Output**: $U, V$, projection matrices.
1: $(P, S, Q) \leftarrow \text{SVD}(G)$
2: $U \leftarrow P[:, : r]$
3: $V \leftarrow Q[:, : r]$
4: **return** $U, V$

---

To retain the advantage of memory efficiency of zero-order optimizations, we calculate the gradient using the MeZO (Malladi et al., 2023) method, as shown in lines 5-6 of Algorithm 3. Before calculating the projection matrix each time, the gradient calculation is required. Then, as shown in the algorithm 1, the $U$ and $V$ matrices are updated according to the gradient obtained this time. We use the SVD method to calculate the projection matrix. Through this method, the original gradient is projected onto a compact space $R \in \mathbb{R}^{r \times r}$: $R = U^T G V$. After that, we can generate a low-rank perturbation $Z$ in this space, as shown in

---

**Algorithm 2** PerturbParams($W, \mathcal{U}, \mathcal{V}, r, \varepsilon, s$)

**Input**: i) $W$, model parameter set; ii) $\mathcal{U}$ and $\mathcal{V}$, projection matrix sets; iii) $r$, rank; iv) $\varepsilon$, perturbation scale; v) $s$, seed.
**Output**: Model parameter set after perturbation.
1: ResetGenerator($s$)
2: **for** $i = 1, 2, \ldots, l$ **do**
3: $\quad Z_i \leftarrow \text{GeneratePerturbMatrix}(r)$
4: $\quad W_i \leftarrow W_i + \varepsilon U_i Z_i V_i^T$
5: **end for**
6: **return** $W$

---

lines 3-4 of the Algorithm 2, and then use the previously calculated $U$ and $V$ matrices to restore this low-rank perturbation to the original high-rank space. In this way, we can successfully apply gradient-based low-rank perturbations to the parameters, and this process introduces no additional overhead compared to the traditional ZO method (i.e., MeZO).

### 4.2 PERIODICAL SUBSPACE UPDATE

As mentioned above, we obtain the gradient using the MeZO (Malladi et al., 2023) method and then calculate the projection matrices $U$ and $V$ via SVD. These two projection matrices jointly determine the gradient approximation and the parameter update of the $t^{th}$ step. However, this iterative update method presents a critical trade-off between computational efficiency and subspace adaptability. High-frequency updates restrict the complete evolution of the gradient subspace while incurring substantial computational costs, particularly due to the need for gradient recomputation before each projection matrix update. In contrast, low-frequency updates may fail to capture the dynamic variations in the gradient subspace throughout the training process.

Therefore, we propose a periodic subspace update strategy. As presented in lines 4-10 in Algorithm 3, we use the MeZO method to calculate the gradient once at the start step and every $F$ steps thereafter. Then the obtained gradient is used to update the projection matrices $U$ and $V$, and keep them unchanged in the subsequent steps. We have experimentally proved the effectiveness and necessity of this strategy. As shown in Table 4, the appropriate update frequency can not only ensure efficiency but also bring significant improvements to model performance.

---

**Algorithm 3** SVD-0

---

**Input**: i) $W_i \in \mathbb{R}^{m_i \times n_i}, i = 1, \ldots, l$, parameter matrix in the $i$-th layer; ii) $\mathcal{L}$, loss; iii) $T$, step budget; iv) $\epsilon$, perturbation scale; v) $\{\eta^t\}$, learning rate schedule; vi) $F$, subspace update frequency; vii) $r$, rank.

1: **for** $t = 1, \ldots, T$ in parallel **do**
2: $\quad \mathcal{B}^t \leftarrow \text{SampleMinbatch}(s^t)$ {Sample a minibatch $\mathcal{B}^t \subset \mathcal{D}$ and a random seed $s^t$}
3: $\quad$ **for** $i = 1, 2, \ldots, l$ **do**
4: $\quad\quad$ **if** $t \mod F \equiv 0$ **then**
5: $\quad\quad\quad G_i \leftarrow \text{EstimateGradient}(W_i^t, \epsilon)$ {Estimate the gradient of $W_i^t$ using MeZO}
6: $\quad\quad\quad U_i^t, V_i^t \leftarrow \text{GenerateProjMatrix}(G_i, r)$
7: $\quad\quad$ **else**
8: $\quad\quad\quad U_i^t \leftarrow U_i^{t-1}, V_i^t \leftarrow V_i^{t-1}$
9: $\quad\quad$ **end if**
10: $\quad$ **end for**
$\quad\quad \{W^t = \{W_i^t\}_{i=1}^l, \mathcal{U}^t = \{U_i^t\}_{i=1}^l, \mathcal{V}^t = \{V_i^t\}_{i=1}^l\}$
11: $\quad W^t \leftarrow \text{PerturbParams}(W^t, \mathcal{U}^t, \mathcal{V}^t, r, \varepsilon, s^t)$
12: $\quad \ell_+^t \leftarrow \mathcal{L}(W^t; \mathcal{B}^t)$
13: $\quad W^t \leftarrow \text{PerturbParams}(W^t, \mathcal{U}^t, \mathcal{V}^t, r, -2\varepsilon, s^t)$
14: $\quad \ell_-^t \leftarrow \mathcal{L}(W^t; \mathcal{B}^t)$
15: $\quad W^t \leftarrow \text{PerturbParams}(W^t, \mathcal{U}^t, \mathcal{V}^t, r, \varepsilon, s^t)$
16: $\quad \rho^t \leftarrow (\ell_+^t - \ell_-^t)/(2\varepsilon)$
17: $\quad \text{ResetGenerator}(s^t)$ {Reset random number generator with seed $s^t$}
18: $\quad$ **for** $i = 1, 2, \ldots, l$ **do**
19: $\quad\quad Z_i^t \leftarrow \text{GeneratePerturbMatrix}(r)$ {Regenerate the perturbation matrix $Z_i^t \in \mathbb{R}^{r \times r}$ whose entries are sampled from $\mathcal{N}(0, 1)$}
20: $\quad\quad W_i^{t+1} \leftarrow W_i^t - \eta^t \rho^t \left( U_i^t Z_i^t V_i^{t\mathsf{T}} \right)$
21: $\quad$ **end for**
22: **end for**

---

Table 1: Computational cost (in minutes) comparison.

| Method | WIC | ReCoRD | FiQA-SA | TFNS | MultiRC |
|---|---|---|---|---|---|
| MeZO (Malladi et al., 2023) | 114.7 | 211.4 | 71.5 | 113.3 | 1125.3 |
| SubZero (Yu et al., 2024) | 114.5 | 207.5 | 71.3 | 124.9 | 1149.5 |
| SVD-0 | 116.1 | 220.6 | 76.3 | 116.2 | 1154.8 |

Table 2: Memory cost comparison.

| Method | RoBERTa-large | OPT-1.3B | OPT-13B |
|---|---|---|---|
| MeZO (Malladi et al., 2023) | 2.042GB | 4.732GB | 27.693GB |
| LOZO (Chen et al., 2025) | 2.042GB | 4.732GB | 27.789GB |
| SVD-0 | 2.562GB | 4.891GB | 28.767GB |

As shown in Table 1, we compared two representative ZO variants (i.e., MeZO (Malladi et al., 2023) and SubZero (Yu et al., 2024)) and our SVD-0 method. The WIC (Pilehvar & Camacho-Collados, 2019) and ReCoRD (Zhang et al., 2018) datasets were paired with the OPT-1.3B model, the FiQA-SA (Maia et al., 2018) and TFNS (Magic, 2022) datasets were paired with the Qwen-1.8B model, whereas the MultiRC (Khashabi et al., 2018) dataset was paired with the OPT-13B model. Each model was fine-tuned on its respective task for 20,000 steps. The findings show that SVD-0 requires a marginally longer training period, approximately 7% longer than the two ZO variants. It's worth mentioning that the time complexity for SVD processes remains at $O(n^3)$, where $n$ is the matrix's dimension, ensuring that the upper bound of the training time complexity remains unchanged. In practice, the extra time required by SVD operations is negligible compared to the benefits gained in classification, multiple-choice, and generation tasks.

Despite reducing computational complexity, this strategy will result in minimal additional memory usage, as shown in Table 2. We adopt a layer-wise parameter update strategy, where we update only the parameters of a specific layer of the model simultaneously. This means that during the entire training process, we only need to store two additional small matrices at the same time, including the projection matrices $U \in \mathbb{R}^{m \times r}$ and $V \in \mathbb{R}^{n \times r}$, where $r$ is much smaller than the dimension

of the parameter matrix $\boldsymbol{\theta} \in \mathbb{R}^{m \times n}$. Therefore, the memory usage introduced by the two matrices remains at the same low level as that introduced in (Yu et al., 2024). This strategy makes our method almost consistent with the memory required by the MeZO (Malladi et al., 2023) method without any performance loss, and maintains the memory-saving advantage of the ZO method.

## 5 CONVERGENCE ANALYSIS

In this section, we analyze the convergence of our proposed SVD-0. Following the derivations in (Yu et al., 2024; Nozawa et al., 2025) and (Zhao et al., 2024a), we first present our proposition along with the corresponding lemma.

**Lemma 1.** *(**Low-rank subspace of weight matrices** (Zhao et al., 2024a)). Gradient matrices become low-rank during fine-tuning. The weight matrix update can be formed as:*

$$\boldsymbol{\theta}_T = \boldsymbol{\theta}_0 + \eta \sum_{t=0}^{T-1} \widetilde{\nabla} f(\boldsymbol{\theta}_t), \quad \widetilde{\nabla} f(\boldsymbol{\theta}_t) = \boldsymbol{U}_t(\boldsymbol{U}_t^{\top} \nabla f(\boldsymbol{\theta}_t) \boldsymbol{V}_t) \boldsymbol{V}_t^{\top}, \tag{3}$$

*where $\eta$ is the learning rate, $\boldsymbol{U}_t \in \mathbb{R}^{m \times r}$ and $\boldsymbol{V}_t \in \mathbb{R}^{n \times r}$ are projection matrices and can be approximated by the spectrum of $\nabla f(\boldsymbol{\theta}_t)$ through $(\boldsymbol{U}, \boldsymbol{V}) = SVD(\nabla f(\boldsymbol{\theta}_t))$.*

Lemma 1 shows that subspace projection matrices can be approximated by adopting SVD on gradients. Given that the SPSA is an unbiased approximation of the exact gradient $\nabla f(\boldsymbol{\theta})$, we can use the SPSA gradient to compute the two projection matrices.

**Proposition 1.** *(**Block-diagonal matrix based on SVD**). The singular matrices $\boldsymbol{U}$ and $\boldsymbol{V}$ are column-orthogonal. Therefore, we can similarly define the following notations based on Equation 1:*

$$\boldsymbol{P} = \text{bdiag}(\boldsymbol{V}_1 \otimes \boldsymbol{U}_1, \dots, \boldsymbol{V}_l \otimes \boldsymbol{U}_l),$$

$$\boldsymbol{z} = \left[ \text{vec}(\boldsymbol{Z}_1)^{\top}, \dots, \text{vec}(\boldsymbol{Z}_l)^{\top} \right]^{\top}, \tilde{\boldsymbol{z}} = \left[ \text{vec}(\tilde{\boldsymbol{Z}}_1)^{\top}, \dots, \text{vec}(\tilde{\boldsymbol{Z}}_l)^{\top} \right]^{\top}.$$

Proposition 1 indicates that the projection matrices in our method exhibit the same properties as the column-orthogonal matrices discussed in (Yu et al., 2024). Consequently, the subsequent theoretical analysis can follow the same approach as that demonstrated in (Yu et al., 2024).

**Lemma 2.** *(**Bounded gradient estimation error** (Yu et al., 2024)). For the gradient estimation in Equation 2, the following two properties hold.*

*i) By using gradient estimation in Equation 2, the estimated gradient $\widehat{\nabla} f(\boldsymbol{\theta})$ is equivalent to:*

$$\widehat{\nabla} f(\boldsymbol{\theta}) = \frac{f(\boldsymbol{\theta} + \varepsilon \boldsymbol{P} \boldsymbol{z}) - f(\boldsymbol{\theta} - \varepsilon \boldsymbol{P} \boldsymbol{z})}{2\varepsilon} \boldsymbol{P} \boldsymbol{z}, \tag{4}$$

*where $\boldsymbol{z} \sim \mathcal{N}(\boldsymbol{0}, \boldsymbol{I}_q)$, $\varepsilon > 0$, $\boldsymbol{P} \in \mathbb{R}^{d \times q}$ satisfies $\boldsymbol{P}^{\top} \boldsymbol{P} = \boldsymbol{I}_q$ with $d = \sum_{i=1}^{l} m_i n_i$ and $q = lr^2$.*

*ii) Let $\boldsymbol{z} \sim \mathcal{N}(\boldsymbol{0}, \boldsymbol{I}_q)$, and $f \in C_{L_2}^{2,2}(\mathbb{R}^d)$. Based on Equation 4 whose properties have been analyzed in (Nozawa et al., 2025), our method has the same bounded gradient estimation error as that in (Yu et al., 2024):*

$$\left\| \mathbb{E}_{\boldsymbol{z}} \left[ \widehat{\nabla} f(\boldsymbol{\theta}) \right] - \boldsymbol{P} \boldsymbol{P}^{\top} \nabla f(\boldsymbol{\theta}) \right\|_2 \leq \frac{\varepsilon^2}{6} L_2 (q + 4)^2. \tag{5}$$

*Note that $f \in C_L^{s,p}(S)$ denotes the class of s-th smooth and p-th L-smooth functions over the set S.*

**Theorem 1.** *(**Convergence of SVD-0**). Consider the optimization problem $\boldsymbol{x}^* = \underset{\boldsymbol{x} \in \mathbb{R}^d}{\arg \min} f(\boldsymbol{x})$, in which $f \in C_{L_1}^{1,1}(\mathbb{R}^d)$ and $f$ exhibits non-convex behavior. Define the stochastic sequence $\mathcal{E}_k = (\boldsymbol{z}_0, \boldsymbol{z}_1, \dots, \boldsymbol{z}_k)$, where each $\boldsymbol{z}_k$ follows the normal distribution $\mathcal{N}(\boldsymbol{0}, \boldsymbol{I}_q)$. Set the step-size parameter as $\eta = \frac{1}{4(q+4)L_1}$. Let $\{\boldsymbol{x}_k\}_{k>0}$ denote the iterates produced via Algorithm 3. For SVD-0, we establish its convergence rate as:*

$$\frac{1}{T} \sum_{k=0}^{T-1} \mathbb{E}_{\mathcal{E}_k} \left[ \left\| \nabla f(\boldsymbol{x}_k) \right\|^2 \right] \leq \epsilon,$$

*Under the scaling $T = \Omega\left(\dfrac{d}{\epsilon^2}\right)$ for $\varepsilon \leq \mathcal{O}\left(\dfrac{1}{q^{3/2}d^{1/2}L_1^{3/2}}\right)$, this aligns with prior theoretical derivations.*

By combining Proposition 1 and Lemma 2 within the framework proposed in (Yu et al., 2024), Theorem 1 demonstrates that our SVD-0 achieves a convergence rate of $\mathcal{O}(\sqrt{\frac{d}{T}})$, matching the rate derived in (Yu et al., 2024). For a more detailed explanation, please see Appendix D.

## 6 EXPERIMENTS

To evaluate the effectiveness of our approach, we implemented SVD-0 using the PyTorch framework (version 20.10). All experiments were conducted on a Linux workstation equipped with CentOS, featuring two NVIDIA A100-40GB GPUs, dual Intel Xeon 6240R CPUs, and 384GB of RAM. The following presents the dataset settings and ZO baselines used in the experiments. Please refer to Appendix A for our detailed model settings.

**Dataset Settings.** For OPT models, we experimented with the SuperGLUE benchmark (Wang et al., 2019), which consists of various types of tasks, including classification tasks (e.g., SST-2 (Socher et al., 2013), RTE (Bar Haim et al., 2006; Bentivogli et al., 2009; Dagan et al., 2005; Giampiccolo et al., 2007), CB (de Marneffe et al., 2019), BoolQ (Clark et al., 2019), WSC (Levesque et al., 2012), WIC (Pilehvar & Camacho-Collados, 2019), and MultiRC (Khashabi et al., 2018)), multiple choice tasks (e.g., COPA (Roemmele et al., 2011) and ReCoRD (Zhang et al., 2018)), and generation tasks (e.g., SQuAD (Rajpurkar et al., 2016) and DROP (Dua et al., 2019)). Here, for each task, we randomly selected 1000 samples for training, 500 samples for validation, and 1000 samples for testing. For the RoBERTa-large model, in addition to the task SST-2, we investigated three more tasks, i.e., SST-5 (Socher et al., 2013), SNLI (Bowman et al., 2015), and MNLI (Williams et al., 2018). In this case, we fixed the parameter $k$ at 512 throughout the training and validation phases, indicating that 512 samples are allocated for each category. For the testing phase, we randomly chose a total of 1000 samples.

**ZO Baselines.** Our SVD-0 method was evaluated against six latest ZO optimization algorithms, i.e., MeZO (Malladi et al., 2023), ZO-AdaMU (Jiang et al., 2024), S-MeZO (Liu et al., 2024), SubZero (Yu et al., 2024), LOZO (Chen et al., 2025), and HiZOO (Zhao et al., 2024b). Meanwhile, we examined three memory-efficient inference-only approaches, i.e., zero-shot evaluation, in-context learning (ICL) (Brown et al., 2020), and linear probing (LP) (Kumar et al., 2022).

We designed our experiments to explore the following research questions (RQs).
**RQ1 (Superiority of SVD-0)**: To what extent does SVD-0 outperform SOTA methods in accuracy?
**RQ2 (Impact of Hyperparameters)**: What are the impacts of critical hyperparameters (e.g., learning rate, subspace rank, subspace update frequency) on SVD-0-based fine-tuning?
**RQ3 (Applicability of SVD-0)**: How does SVD-0 perform when fine-tuning models of varying sizes or architectures (e.g., masked or causal language models)?

### 6.1 COMPARISON WITH STATE-OF-THE-ARTS (RQ1)

We compared our proposed SVD-0 method with the SOTA ZO optimizers. The experiments were conducted on the SuperGLUE benchmark employing both the OPT-13B and OPT-1.3B language models of different sizes. Note that in each experiment, we applied the adopted stochastic gradient descent (SGD) or ZO method to all model parameters.

Table 3 compares the fine-tuning performance of the OPT-13B model on SuperGLUE benchmark tasks. Here, we considered three types of fine-tuning methods: i) the traditional fine-tuning method (i.e., SGD) with backpropagation; ii) inference-only methods (i.e., Zero-shot, ICL, and LP) without fine-tuning; and iii) memory-efficient ZO-based methods. To enable a fair comparison between ZO-based methods, we used the MeZO method here as a reference. We evaluated the overall performance across each classification task category and denoted the improvement in performance compared to the baseline (i.e., MeZO) in the sub-column labeled "Total". For example, the total performance on multiple choice tasks with MeZO and SVD-0 is 169.0 and 171.2, respectively. In this case, SVD-0 improves inference performance by 1.30% compared to MeZO. From the results provided in

Table 3: Comparison of OPT-13B fine-tuning performance (%) on SuperGLUE, where the best results are presented in **bold** and the second-best results are highlighted with underlines.

| Method | Classification Task | | | | | | | | Multiple Choice Task | | | Generation Task | | | All Task |
|---|---|---|---|---|---|---|---|---|---|---|---|---|---|---|---|
| | SST-2 | RTE | CB | BoolQ | WSC | WIC | MultiRC | Total | COPA | ReCoRD | Total | SQuAD | DROP | Total | Total |
| SGD | 94.9 | 82.3 | 85.7 | 78.4 | 65.3 | 65.8 | 74.2 | - | 90.0 | 82.4 | - | 88.0 | 35.5 | - | - |
| Zero-shot | 58.8 | 59.6 | 46.4 | 59.0 | 38.5 | 55.0 | 46.9 | - | 80.0 | 81.2 | - | 46.2 | 14.6 | - | - |
| ICL (Brown et al., 2020) | 87.0 | 62.1 | 57.1 | 66.9 | 39.4 | 50.5 | 53.1 | - | 87.0 | 82.5 | - | 75.9 | 29.6 | - | - |
| LP (Kumar et al., 2022) | 93.4 | 68.6 | 67.9 | 59.3 | 63.5 | 60.2 | 63.5 | - | 55.0 | 27.1 | - | 3.7 | 11.1 | - | - |
| MeZO (Malladi et al., 2023) | 92.1 | 71.5 | 71.4 | 74.4 | 61.5 | 60.0 | 60.1 | 0% | 87.0 | 82.0 | 0% | 84.2 | 31.2 | 0% | 0% |
| ZO-AdaMU (Jiang et al., 2024) | 92.1 | 72.9 | 67.9 | 73.0 | 61.5 | 60.7 | 63.0 | 0.02% | **89.0** | **83.0** | **1.78%** | 82.4 | **32.0** | -0.87% | 0.27% |
| S-MeZO (Liu et al., 2024) | 92.3 | **76.9** | **75.0** | **76.5** | 61.1 | 58.2 | **63.3** | 2.51% | 87.0 | 71.2 | -6.39% | 77.9 | 31.9 | -4.85% | -0.53% |
| HiZOO (Zhao et al., 2024b) | 91.3 | 69.3 | 69.4 | 67.3 | 63.5 | 59.4 | 55.5 | -3.12% | 88.0 | 81.4 | 0.24% | 81.9 | 31.3 | -1.91% | -2.21% |
| LOZO (Chen et al., 2025) | 91.7 | 70.4 | 69.6 | 71.9 | 63.5 | 60.8 | 63.0 | -0.02% | **89.0** | 81.3 | 0.77% | 84.9 | 30.7 | 0.17% | 0.18% |
| SubZero (Yu et al., 2024) | 92.1 | 74.0 | 73.2 | 75.3 | **65.4** | 60.8 | 61.0 | 2.20% | 88.0 | 82.3 | 0.77% | 84.5 | **32.0** | **0.95%** | 1.70% |
| SVD-0 | **93.6** | 75.5 | 71.4 | 75.2 | 63.5 | **65.4** | 60.6 | **2.89%** | **89.0** | 82.2 | 1.30% | **85.1** | 30.9 | 0.52% | **2.19%** |

the "Total" sub-columns, we can find that SVD-0 can always achieve top-2 inference performance. Furthermore, we used the final column to show the relative performance improvement for all tasks. From this column, we can find that SVD-0 achieves the best overall performance. Interestingly, while S-MeZO matches SVD-0 in the number of tasks where it excels, its overall performance, shown in the final column, is noticeably inferior to SVD-0 and even falls short of the reference (i.e., MeZO).

## 6.2 IMPACTS OF HYPERPARAMETERS (RQ2)

In this experiment, we investigate three key hyperparameters (i.e., subspace update frequency, rank, and learning rate) to evaluate their impacts on fine-tuning performance.

Table 4: Impact of subspace update frequency, where the best results are highlighted in **bold**.

| Frequency | SST-2 | RTE | CB | BoolQ | WSC | WIC | MultiRC | COPA | ReCoRD | SQuAD | DROP |
|---|---|---|---|---|---|---|---|---|---|---|---|
| 50 | 90.5 | 57.0 | 64.3 | 65.0 | 63.5 | 55.6 | 57.5 | 72.0 | **72.4** | 74.2 | 23.0 |
| 500 | 89.5 | 55.6 | 69.6 | 64.1 | **63.5** | 53.9 | 58.1 | **73.0** | 72.2 | **74.3** | 22.9 |
| 1000 | **90.6** | **58.5** | 71.4 | 65.2 | **63.5** | 56.4 | 58.2 | **73.0** | 72.1 | 73.7 | **24.0** |
| 2000 | 89.2 | 56.7 | **73.2** | 64.5 | 62.5 | 57.4 | 58.1 | **73.0** | 71.7 | 72.6 | 23.8 |
| 20000 | 89.8 | 56.3 | 71.4 | **65.3** | 62.5 | **57.5** | **58.2** | 72.0 | 72.1 | 72.6 | 22.6 |

For the subspace update frequency $F$, our goal is to evaluate the impact of varying this frequency on model performance across different tasks. We conducted experiments based on the OPT-1.3B model, with a fixed rank of $r = 24$ and a learning rate of $1e-7$. In this analysis, we evaluated five frequencies at varying magnitudes, specifically selected from the set $\{50, 500, 1000, 2000, 20000\}$. Table 4 provides the experimental results. From this table, we can find that when the frequency is set to 1000 (i.e., the subspace is updated every 1000 steps), SVD-0 achieves the best performance in six of the eleven tasks. Note that SVD-0-based fine-tuning is not sensitive to the hyperparameter $F$. Therefore, we suggest setting $F$ to 1000 by default for fine-tuning.

We investigated the rank of hyperspace (i.e., $r$) and the learning rate in tandem. Table 5 presents the fine-tuning performance under various combinations of these two hyperparameters, where the rank is selected from $\{2, 24, 48, 64, 128\}$ and the learning rate is selected from $\{1e-7, 5e-7, 1e-6\}$. All the experimental results are collected based on the SST-2 task using the OPT-1.3B model, with a fixed subspace update frequency of 1000. This table shows that the fine-tuning performance is weak when the rank is low (i.e., $r = 2$). While elevating the rank can enhance fine-tuning performance, the

Table 5: Impacts of rank and learning rate on inference.

| Rank\LR | 1e−7 | 5e−7 | 1e−6 |
|---|---|---|---|
| 2 | 87.7 | 91.2 | 86.7 |
| 24 | 90.6 | 92.2 | 90.3 |
| 48 | 89.5 | 91.6 | 90.1 |
| 64 | 89.9 | 90.4 | 91.6 |
| 128 | 90.0 | 91.3 | 90.6 |

extent of this enhancement becomes negligible once the rank surpasses 24. At low ranks, the performance can vary significantly with different learning rates. In contrast, increasing rank tends to reduce this variability in performance. Moreover, we observe a similar trend for the learning rate hyperparameter, where setting the learning rate to $5e-7$ achieves the best performance for most rank settings. However, increasing the learning rates can lead to a decline in inference performance.

## 6.3 IMPACT OF MODEL SIZES AND ARCHITECTURES (RQ3)

In Table 3, we have evaluated the adaptability of SVD-0 to large-scale LLMs. To further validate the generalizability of our approach, we extended our evaluation to the OPT-1.3B model, using

representative tasks of different types. These tasks include SST-2 and WIC, which are classification tasks, ReCoRD, a multiple-choice task, and SQuAD, a generation task. Table 6 presents the results of the comparison between four ZO-based fine-tuning methods, where the last column shows the average fine-tuning performance of the four tasks. According to this table, we can see that SVD-0 is also well-suited for fine-tuning on small-scale LLMs. Although LOZO delivers the highest performance in this experiment, the difference in average fine-tuning performance between SVD-0 and LOZO is minimal (i.e., only 0.2%). Note that SVD-0 achieves better performance than MeZO, the reference method, while SubZero fails to beat MeZO. Moreover, SVD-0 can consistently outperform its counterpart (i.e., SubZero) by an average of 0.7%. All these observations substantiate the efficiency of our method in enhancing subspaces for optimizing LLMs.

Table 6: Fine-tuning performance (%) comparison for OPT-1.3B, where the top-2 results are marked in bold and with underlines, respectively.

| Method | SST-2 | WIC | ReCoRD | SQuAD | AVG. |
|---|---|---|---|---|---|
| MeZO (Malladi et al., 2023) | 91.7 | 61.1 | 72.2 | 77.4 | 75.6 |
| LOZO (Chen et al., 2025) | **93.2** | **62.4** | 71.9 | **78.1** | **76.4** |
| SubZero (Yu et al., 2024) | 91.9 | 60.7 | 72.0 | 77.6 | 75.5 |
| SVD-0 (Ours) | 93.0 | 61.1 | **73.0** | 77.6 | 76.2 |

Table 7: Fine-tuning performance (%) comparison for RoBERTa-large, where the top-2 results are marked in bold and with underlines, respectively.

| Method | SST-2 | SST-5 | SNLI | MNLI |
|---|---|---|---|---|
| Zero-shot | 79.0 | 35.5 | 50.2 | 48.8 |
| MeZO (Malladi et al., 2023) | 93.7 (0.4) | 53.9 (1.9) | 84.8 (1.1) | 76.6 (0.8) |
| LOZO (Chen et al., 2025) | 94.1 (0.7) | 53.0 (0.4) | **85.4 (0.8)** | **80.4 (1.0)** |
| SVD-0 (Ours) | **94.4 (0.7)** | **54.4 (0.7)** | 85.4 (1.3) | 80.4 (1.5) |

We investigated the fine-tuning performance of different optimization methods on RoBERTa-large, where we considered four downstream tasks, including two sentiment classification tasks (i.e., SST-2 and SST-5) and two natural language inference tasks (i.e., SNLI and MNLI). For a fair comparison, like the work in (Chen et al., 2025), we performed fine-tuning on each task five times using different random seeds. Table 7 presents the experimental results, reflecting both the average inference performance and its standard deviation (indicated in parentheses) for each combination of fine-tuning methods and tasks. From this table, we can see that SVD-0 performs the best compared to SOTA ZO optimization methods, demonstrating the adaptability of our approach to various model architectures.

Table 8: Fine-tuning performance (%) comparison for Qwen-1.8B, where the top-2 results are marked in bold and with underlines, respectively.

| Method | SST-2 | WIC | ReCoRD | Total |
|---|---|---|---|---|
| MeZO (Malladi et al., 2023) | 78.3 | 55.6 | 64.8 | 0% |
| LOZO (Chen et al., 2025) | 81.7 | 55.2 | 64.8 | 1.51% |
| SubZero (Yu et al., 2024) | 80.8 | 56.7 | 65.2 | 2.01% |
| SVD-0 | **82.2** | **57.2** | **65.3** | **3.02%** |

Table 9: Fine-tuning performance (%) comparison for OPT-1.3B on financial datasets, where the top-2 results are marked in bold and with underlines, respectively.

| Method | FPB | FIQA-SA | TFNS | NWGI | Total |
|---|---|---|---|---|---|
| MeZO (Malladi et al., 2023) | 65.3 | 81.4 | 74.7 | 48.5 | 0% |
| LOZO (Chen et al., 2025) | 61.3 | **85.1** | 71.6 | **53.7** | 0.67% |
| SubZero (Yu et al., 2024) | 66.4 | 84.0 | **78.4** | 49.7 | 3.19% |
| SVD-0 | **74.1** | 84.0 | 76.3 | 52.8 | **6.41%** |

To further validate the generalization ability of our method on cutting-edge models, we conducted experiments based on the Qwen-1.8B model. We exclusively compared our method against the baseline (i.e., MeZO (Malladi et al., 2023)) and the two most recent ZO baseline techniques (i.e., LOZO (Chen et al., 2025) and SubZero (Yu et al., 2024)). Table 8 shows that our approach still achieves the best performance, indicating the adaptability and generalizability of our method in cutting-edge models. Moreover, we assessed SVD-0 on datasets derived from four financial sentiment analysis benchmarks, including FPB (Malo et al., 2014), FIQA-SA (Maia et al., 2018), TFNS (Magic, 2022), and NWGI (Yang, 2023). As shown in Table 9, SVD-0 achieves the highest total performance, demonstrating the method's reliability and efficiency across various domains and task types.

## 6.4 DISCUSSION

**Limitations.** While the SVD-0 technique improves the ZO subspace fine-tuning approach, the accuracy of the subspace projection matrices is significantly influenced by the precision of the ZO gradients. In smaller models, such as the OPT-1.3B, the ZO gradients may have a greater approximation error, which can result in decreased precision in obtaining the projection matrices.

**Border Impacts.** In this paper, we introduced a new approach to derive more precise projection matrices, which can be used to improve the effectiveness of ZO subspace fine-tuning techniques for LLMs. Our method utilizes SVD on ZO gradients to extract projection matrices, eliminating the need for the memory-demanding FO gradients. Our theoretical convergence analysis, in conjunction with the experimental findings, demonstrates that our research makes a positive contribution to the advancement of memory-efficient fine-tuning methods for LLMs.

## 7 CONCLUSION

Although various zeroth-order (ZO) optimization methods have been proposed to enable memory-efficient fine-tuning for large language models (LLMs), due to the use of random subspaces, most of them suffer from inaccurate gradient estimation, resulting in inferior training performance. To address this problem, this paper presents a novel ZO subspace fine-tuning method named SVD-0. By precisely capturing fine-tuning subspaces, SVD-0 enables the construction of projection matrices with higher accuracy, thereby achieving more accurate gradient estimation and improving the LLM fine-tuning performance. Extensive experimental findings demonstrate the efficacy of SVD-0 in dealing with complex language modeling tasks. In the future, we plan to integrate our SVD-0 method with parameter quantization techniques to reduce the memory requirements of LLM fine-tuning.

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

# A DETAILED EXPERIMENTAL SETTINGS

## A.1 MODEL SETTINGS

In our experiments, we considered both large-scale autoregressive language models (i.e., OPT-1.3B and OPT-13B (Zhang et al., 2022)) and a masked language model (i.e., RoBERTa-large (Liu et al., 2019)). In the experiments, all ZO methods used a batch size of 16, except where specified, since larger batches help minimize the gradient approximation variance. We chose MeZO as the main baseline because it is the first widely adopted ZO optimizer for LLMs, and included the first-order SGD as a reference for optimization. In line with previous research (Malladi et al., 2023; Zhang et al., 2024), our experiments utilized standardized prompt templates, which are crucial in influencing the performance of ZO methods. Moreover, to ensure a fair comparison, we considered multiple values for each key hyperparameter. For example, we investigated the following hyperparameter configurations for OPT-13B: a learning rate in $\{1e{-}7, 2e{-}7, 5e{-}7, 1e{-}6\}$, $\epsilon = 1e - 3$, a batch size of 16 (except for MultiRC and DROP which have a batch size of 8), a rank in $\{24, 32, 48, 64, 128\}$,

and a subspace update frequency in $\{500, 1000, 2000\}$. Please refer to Appendix A for detailed configurations of other models. Similar to the work in (Yu et al., 2024), we conducted an exhaustive grid search over hyperparameters for each pairing of ZO methods and LLMs, using the best results for an equitable comparison.

## A.2 DATASET SETTINGS.

For OPT models, we experimented with the SuperGLUE benchmark (Wang et al., 2019), which consists of various types of tasks, including classification tasks (e.g., SST-2 (Socher et al., 2013), RTE (Bar Haim et al., 2006; Bentivogli et al., 2009; Dagan et al., 2005; Giampiccolo et al., 2007), CB (de Marneffe et al., 2019), BoolQ (Clark et al., 2019), WSC (Levesque et al., 2012), and WIC (Pilehvar & Camacho-Collados, 2019)), multiple choice tasks (e.g., COPA (Roemmele et al., 2011) and ReCoRD (Zhang et al., 2018)), and generation tasks (e.g., SQuAD (Rajpurkar et al., 2016) and DROP (Dua et al., 2019)). Here, for each task, we randomly selected 1000 samples for training, 500 samples for validation, and 1000 samples for testing. For the RoBERTa-large model, in addition to the task SST-2, we investigated three more tasks, i.e., SST-5 (Socher et al., 2013), SNLI (Bowman et al., 2015), and MNLI (Williams et al., 2018). In this case, we fixed the parameter $k$ at 512 throughout the training and validation phases, indicating that 512 samples are allocated for each category. For the testing phase, we randomly chose a total of 1000 samples.

## A.3 HYPERPARAMETER SETTINGS

This section provides a detailed overview of the hyperparameters employed in our grid search across the experiments, as depicted in Tables 10 and 12. For the OPT model, we carried out 20,000 steps for each method. Both the SGD and ZO methodologies were implemented for an identical number of steps. In the remaining RoBERTa experiments, ZO optimization strategies were applied over 100,000 training steps. For both models, we evaluated the validation loss every 1,000 training steps to determine the optimal model checkpoint. In the S-MeZO strategy, the sparsity rate is set to 0.75.

Table 10: The hyperparameter grids used for OPT-13B experiments.

| Method | Hyperparameters | | | | |
|---|---|---|---|---|---|
| | Batch Size | Learning Rate | $\epsilon$ | Rank | Update Interval |
| SGD | 16 | $\{1e-4, 1e-3, 5e-3\}$ | – | – | – |
| MeZO (Malladi et al., 2023) | 16 | $\{1e-7, 2e-7, 5e-7, 1e-6\}$ | $1e-3$ | – | – |
| S-MeZO (Liu et al., 2024) | 16 | $\{1e-6, 5e-6\}$ | $1e-3$ | – | – |
| LOZO (Chen et al., 2025) | 16 | $\{1e-7, 1e-6\}$ | $\{1e-3, 1e-4\}$ | $\{1, 2, 4\}$ | $\{50, 100\}$ |
| SubZero (Yu et al., 2024) | 16 | $\{1e-7, 2e-7, 5e-7, 1e-6\}$ | $1e-3$ | $\{32, 64, 128, 256\}$ | $\{500, 1000, 2000\}$ |
| SVD-0 | 16 | $\{1e-7, 2e-7, 5e-7, 1e-6\}$ | $1e-3$ | $\{24, 32, 48, 64, 128\}$ | $\{500, 1000, 2000\}$ |

Table 11: The hyperparameter grids used for OPT-1.3B experiments.

| Method | Hyperparameters | | | | |
|---|---|---|---|---|---|
| | Batch Size | Learning Rate | $\epsilon$ | Rank | Update Interval |
| MeZO (Malladi et al., 2023) | 16 | $\{1e-7, 5e-7, 1e-6\}$ | $1e-3$ | – | – |
| LOZO (Chen et al., 2025) | 16 | $\{1e-7, 1e-6\}$ | $\{1e-3, 1e-4\}$ | $\{1, 2, 4\}$ | $\{50, 100\}$ |
| SubZero (Yu et al., 2024) | 16 | $\{1e-7, 5e-7, 1e-6\}$ | $1e-3$ | $\{24, 48\}$ | $1000$ |
| SVD-0 | 16 | $\{1e-7, 5e-7, 1e-6\}$ | $1e-3$ | $\{8, 24, 48\}$ | $\{50, 500, 1000\}$ |

For all previously mentioned ZO methods, we utilized a consistent learning rate schedule and set the weight decay to zero. Typically, we chose a batch size of 16 for the OPT-1.3B and OPT-13B models across various tasks. Nonetheless, due to limited GPU resources, we reduced the batch size to 8 for the DROP, MultiRC, and SQuAD evaluations.

Table 12: The hyperparameter grids used for RoBERTa-large experiments.

| Method | Hyperparameters | | | | |
|---|---|---|---|---|---|
| | Batch Size | Learning Rate | $\epsilon$ | Rank | Update Interval |
| MeZO (Malladi et al., 2023) | 64 | $\{1e-7, 1e-6, 1e-5\}$ | $1e-3$ | – | – |
| LOZO (Chen et al., 2025) | 64 | $2e-7$ | $1e-3$ | $\{4, 8\}$ | $\{50, 100\}$ |
| SVD-0 | 64 | $1e-6$ | $1e-3$ | $\{8, 16, 24\}$ | 1000 |

## B  DETAILED PRESTUDY RESULTS

Table 13: Impact of rank and batch size.

| Rank\Batch Size | 1 | 2 | 4 | 8 | 16 |
|---|---|---|---|---|---|
| 8 | 0.87 | 0.86 | 0.50 | 0.72 | 0.67 |
| 24 | 0.85 | 0.84 | 0.45 | 0.70 | 0.63 |
| 48 | 0.85 | 0.84 | 0.45 | 0.70 | 0.63 |

For SVD, we perform truncation by restricting the rank to 24 for the singular vectors. In Figure 1, the x-axis indicates the number of training steps completed during fine-tuning, while the y-axis plots the cosine similarity between the ZO gradient at the current step and the FO gradient, both after applying SVD at each step. It is important to note that this cosine similarity is calculated using only a subset of the parameters, rather than as an average across the entirety of the parameters.

We further explored how similarity changes under various rank and batch size configurations. To do this, we examined the ranks of 8, 24, and 48, paired with batch sizes of 1, 2, 4, 8, and 16. Table 13 summarizes these findings. Our results show that reducing the rank leads to a minor decrease in similarity, though the effect is minimal, and overall similarity remains stable. In contrast, modifying batch size results in more noticeable fluctuations in cosine similarity, but there is no clear trend of increase or decrease. In summary, the similarity remains consistently high in most settings.

Table 14: Comparison between SVD and other dimensionality reduction techniques.

| Method | Mean | Std | Min | Max | Median |
|---|---|---|---|---|---|
| Origin | 0.0000 | 0.0005 | -0.0013 | 0.0012 | 0.0000 |
| **After SVD (ours)** | **0.4249** | **0.0257** | **0.3276** | **0.4774** | **0.4278** |
| PCA | -0.0003 | 0.0044 | -0.0141 | 0.0101 | -0.0004 |
| NMF | 0.2464 | 0.0442 | 0.1097 | 0.3594 | 0.2456 |
| Factor Analysis | 0.0001 | 0.0045 | -0.0124 | 0.0136 | -0.0002 |
| Random Proj | -0.0002 | 0.0045 | -0.0112 | 0.0177 | -0.0001 |
| t-SNE | -0.0009 | 0.0157 | -0.0437 | 0.0567 | -0.0012 |

We conducted experiments to investigate why SVD outperforms other methods for dimensionality reduction. Table 14 presents a comparison between our dimensionality reduction strategy ("After SVD") and other techniques (PCA, NMF, Factor Analysis, Random Projection, and t-SNE) based on the calculated gradients. As shown in Table 14, our method yields the highest mean value (0.4249) among all dimensionality reduction techniques, accompanied by a lower standard deviation (0.0257), which highlights its superior and consistent performance.

## C  ADDITIONAL RESULTS

Figure 3 presents the loss curves of SVD-0 and SubZero during training, where the x-axis represents the wall-clock time. Although SVD-0 takes a bit longer than SubZero for each iteration, it consistently reaches lower training loss from the beginning to the end. This indicates that, overall, SVD-0 converges faster in terms of total training time.

We examined the cosine similarity between gradients obtained from different ZO methods and those from FO. Throughout the fine-tuning process of

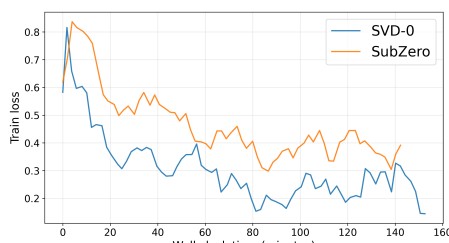

Figure 3: Loss curves plotted against wall-clock time.

the OPT-1.3B model on the SST2 (Socher et al., 2013) dataset, we periodically sampled the actual gradients for a chosen subset of parameters, along with the ZO gradients. The cosine similarity between these ZO and FO gradients was then calculated. The resulting values were averaged across the entire fine-tuning process and normalized to the $[0, 1]$ range to facilitate comparison.

Table 15: Cosine similarity between ZO gradients and the FO gradient.

| Method | Cosine Similarity |
|---|---|
| MeZO (Malladi et al., 2023) | 0 |
| SubZero (Yu et al., 2024) | 0.73 |
| SVD-0 | 1 |

The results presented in Table 15 indicate that SVD-0 achieves the highest level of similarity. This suggests that using SVD to extract the shared features between ZO and FO gradients results in a closer match to the FO gradient. These observations further reinforce the conclusions drawn in our preliminary study.

Table 16: Variance of the estimated gradients.

| Method | Variance |
|---|---|
| MeZO (Malladi et al., 2023) | 5.5 |
| SubZero (Yu et al., 2024) | 5.6 |
| SVD-0 | 5.0 |

We also conducted experiments to evaluate the variance of the estimated gradients. For each approach, we performed tests on the SST-2 dataset with OPT-1.3B using identical hyperparameters. We collected 1,000 samples per method and calculated the variance for the resulting estimated gradients. A summary of the results is provided in Table 16. The findings indicate that SVD-0 yields the lowest variance, further supporting the superiority of SVD-0.

We performed experiments to explore the behavior of SVD-0 when subjected to significantly noisy zero-order (ZO) gradients. Gaussian noise with varying variances was introduced into the ZO gradients for these tests. In particular, we injected zero-mean Gaussian noise into the ZO gradients, with its variance scaled to $K$ times that of the projected gradients, simulating noisy conditions. We then fine-tuned the OPT-1.3B model on the SST-2 (Socher et al., 2013), RTE (Bar Haim et al., 2006; Bentivogli et al., 2009; Dagan et al., 2005; Giampiccolo et al., 2007), and WIC (Pilehvar & Camacho-Collados, 2019) datasets. The results for these experiments are shown in Table 17.

Table 17: Impact of different noise conditions.

| K | SST-2 | RTE | WIC |
|---|---|---|---|
| No noise | 92.2 | 68.2 | 56.3 |
| 0.5 | 87.5 | 68.6 | 56.6 |
| 1 | 88.9 | 66.1 | 56.4 |
| 2 | 87.5 | 66.4 | 55.0 |

In general, increasing the noise variance leads to a modest drop in model performance, yet SVD-0 continues to show resilience in noisy scenarios. Furthermore, the best results are achieved in the absence of noise or when noise levels are minimal, suggesting that SVD-0 effectively leverages the valuable aspects of the ZO gradients to enhance fine-tuning.

To demonstrate that SVD-0 consistently delivers superior performance across various random seeds on OPT, we conducted an experiment where we fine-tuned the OPT-1.3B model on the SST-2 (Socher et al., 2013) dataset. As shown in Table 18, SVD-0 consistently outperforms MeZO (Malladi et al., 2023) across all random seeds.

Table 18: Impact of random seed.

| Method\Seed | 0 | 42 | 123 |
|---|---|---|---|
| MeZO (Malladi et al., 2023) | 90.9 | 88 | 89.2 |
| SVD-0 | **93.0** | **90.1** | **90.4** |

As presented in Table 19, we fine-tuned the OPT-1.3B model on various datasets using the Adam optimizer, applying the same hyperparameters as in the SGD setup. Our results demonstrate that SVD-0, in combination with Adam, consistently surpasses the MeZO baseline. This suggests that SVD-0 is especially effective for real-world fine-tuning scenarios involving the Adam optimizer.

Table 19: Performance on Adam optimizer.

| Method | SST-2 | WIC | RTE |
|---|---|---|---|
| MeZO(Adam) | 75.6 | 55.8 | 54.2 |
| SVD-0(Adam) | 92.9 | 62.4 | 75.5 |

Given that vanilla LoRA uses Adam for fine-tuning, we set up experiments to compare SVD-0 with SGD in fine-tuning scenarios, alongside vanilla LoRA with Adam. All experiments were conducted using the pretrained OPT-1.3B model and evaluated on the SST-2 dataset. Table 20 shows that both SVD-0 (SGD) and LoRA (Adam) deliver similar results on SST-2. However, SVD-0 (SGD) stands out by using much less memory than LoRA (Adam), underscoring its key advantage as a zeroth-order method: substantial memory savings without a loss in performance.

Table 20: Comparison with LoRA.

| Method | Accuracy(%) | Memory(GB) |
|---|---|---|
| LoRA(Adam) | 93.2 | 12.926 |
| SVD-0(SGD) | 93.0 | 4.891 |

We also integrated SVD-0 with LoRA and performed experiments using OPT-13B. We set the learning rates for MeZO (LoRA) and SVD-0 (LoRA) to $\{1.5e{-}5, 3e{-}5, 5e{-}5\}$, the rank of SVD-0 (LoRA) to $\{4, 8, 16\}$, and the subspace change frequency to $\{500, 1000, 2000\}$.

Table 21: Performance of SVD-0 combined with LoRA.

| Method | SST-2 | RTE | CB | BoolQ | WSC | WIC | MultiRC |
|---|---|---|---|---|---|---|---|
| MeZO(LoRA) | 92.2 | 74.4 | 69.6 | 75.2 | 64.4 | 59.7 | 58.2 |
| SVD-0(LoRA) | **93.8** | **76.9** | **71.4** | **76.0** | **65.4** | **61.4** | **59.3** |

As shown in Table 21, SVD-0 (LoRA) consistently achieves superior results compared to MeZO (LoRA) across all datasets, suggesting that integrating SVD-0 with LoRA yields a substantial performance boost.

Table 22: Performance of SVD-0 on various ZO estimators.

| Estimators | MeZO | SPSA | NES |
|---|---|---|---|
| Accuracy(%) | 92.2 | 90.5 | 90.3 |

As presented in Table 22, we replaced MeZO with alternative zeroth-order (ZO) gradient estimators to provide SVD-0 with the necessary gradient information. Specifically, we utilized SPSA and NES (Natural Evolution Strategies) as gradient estimators and conducted experiments on the SST-2 dataset using OPT-1.3B, maintaining consistent hyperparameter settings. The performance outcomes are summarized below. Our results show that, although SPSA and NES perform slightly below MeZO, both methods still deliver robust results. This suggests that SVD-0 is a generally applicable subspace technique that is not dependent on MeZO in particular.

## D PROOFS

Here, we introduce some definitions and lemmas for continuous proofs. The following two lemmas illustrate that the low-rank perturbation matrix for each layer can be represented as a layer-scale projection matrix that is orthogonal across its columns.

**Lemma 3.** *Let* $\tilde{\boldsymbol{Z}} = \boldsymbol{U}\boldsymbol{Z}\boldsymbol{V}^{\mathsf{T}}$, *where* $\boldsymbol{U} \in \mathbb{R}^{m \times r}, \boldsymbol{Z} \in \mathbb{R}^{r \times r}, \boldsymbol{V} \in \mathbb{R}^{n \times r}$, *and* $\boldsymbol{U}^{\mathsf{T}}\boldsymbol{U} = \boldsymbol{V}^{\mathsf{T}}\boldsymbol{V} = \boldsymbol{I}_r$. *Then we have* $\mathrm{vec}(\tilde{\boldsymbol{Z}}) = \boldsymbol{P}\mathrm{vec}(\boldsymbol{Z})$ *and* $\boldsymbol{P}^{\mathsf{T}}\boldsymbol{P} = \boldsymbol{I}_{r^2}$, *where* $\boldsymbol{P} = \boldsymbol{V} \otimes \boldsymbol{U}$.

*Proof.* Since $\mathrm{vec}(\boldsymbol{U}\boldsymbol{Z}\boldsymbol{V}^{\mathsf{T}}) = (\boldsymbol{V} \otimes \boldsymbol{U})\mathrm{vec}(\boldsymbol{Z})$, we only need to show $(\boldsymbol{V} \otimes \boldsymbol{U})^{\mathsf{T}}(\boldsymbol{V} \otimes \boldsymbol{U}) = \boldsymbol{I}_{r^2}$. In fact:

$$(\boldsymbol{V} \otimes \boldsymbol{U})^{\mathsf{T}}(\boldsymbol{V} \otimes \boldsymbol{U}) = (\boldsymbol{V}^{\mathsf{T}} \otimes \boldsymbol{U}^{\mathsf{T}})(\boldsymbol{V} \otimes \boldsymbol{U}) = (\boldsymbol{V}^{\mathsf{T}}\boldsymbol{V}) \otimes (\boldsymbol{U}^{\mathsf{T}}\boldsymbol{U}) = \boldsymbol{I}_r \otimes \boldsymbol{I}_r = \boldsymbol{I}_{r^2}.$$

The proof is completed. $\qquad\square$

We can also show that the low-rank perturbation matrices across all layers can be represented as a model-scale projection matrix.

**Lemma 4.** *Let a block diagonal matrix* $\boldsymbol{P} = \mathrm{bdiag}(\boldsymbol{P}_1, \boldsymbol{P}_2, \cdots, \boldsymbol{P}_l)$ *and* $\tilde{\boldsymbol{z}}_i = \boldsymbol{P}_i \boldsymbol{z}_i$, *where* $\boldsymbol{P}_i^\mathsf{T} \boldsymbol{P}_i = \boldsymbol{I}_{r^2}$ *and* $i = 1, 2, \ldots, l$. *Then we have* $\tilde{\boldsymbol{z}} = \boldsymbol{P} \boldsymbol{z}$, *where* $\tilde{\boldsymbol{z}} = [\tilde{\boldsymbol{z}}_1^\mathsf{T}, \ldots, \tilde{\boldsymbol{z}}_l^\mathsf{T}]^\mathsf{T}$, $\boldsymbol{z} = [\boldsymbol{z}_1^\mathsf{T}, \ldots, \boldsymbol{z}_l^\mathsf{T}]^\mathsf{T}$ *and* $\boldsymbol{P}^\mathsf{T} \boldsymbol{P} = \boldsymbol{I}_{lr^2}$.

*Proof.* It is easy to check that $\tilde{\boldsymbol{z}} = \boldsymbol{P} \boldsymbol{z}$. Besides, we have:

$$\boldsymbol{P}^\mathsf{T} \boldsymbol{P} = \mathrm{bdiag}(\boldsymbol{P}_1^\mathsf{T}, \ldots, \boldsymbol{P}_l^\mathsf{T}) \mathrm{bdiag}(\boldsymbol{P}_1, \ldots, \boldsymbol{P}_l) = \mathrm{bdiag}(\boldsymbol{P}_1^\mathsf{T} \boldsymbol{P}_1, \ldots, \boldsymbol{P}_l^\mathsf{T} \boldsymbol{P}_l) = \boldsymbol{I}_{lr^2}.$$

The proof is completed. $\qquad\square$

According to Lemma 4 and Proposition 1, the perturbation vector of SVD-0 is given by $\tilde{\boldsymbol{z}} = \boldsymbol{P} \boldsymbol{z}$. This is similar to existing random subspace methods, but SVD-0's projection matrix is block diagonal and orthogonal by columns.

**Definition 1.** *We say that the vector* $\boldsymbol{z}$ *is a standard* $n$-*dimensional Gaussian vector, denoted by* $\boldsymbol{z} \sim \mathcal{N}(\boldsymbol{0}, \boldsymbol{I}_n)$, *if its probability density function is given by* $p(\boldsymbol{z}) = \frac{1}{\kappa} e^{-\frac{1}{2}\|\boldsymbol{z}\|^2}$, *where* $\kappa > 0$ *satisfies the condition* $\int_{\mathbb{R}^n} \frac{1}{\kappa} e^{-\frac{1}{2}\|\boldsymbol{z}\|^2} d\boldsymbol{z} = 1$.

**Definition 2.** *Let* $\boldsymbol{z} \sim \mathcal{N}(\boldsymbol{0}, \boldsymbol{I}_n)$. *We say that* $x$ *is a chi-square random variable with* $n$ *degrees of freedom (denoted by* $x \sim \chi^2(n)$*) if* $x = \|\boldsymbol{z}\|^2$.

**Lemma 5.** *Let* $\boldsymbol{z} \sim \mathcal{N}(\boldsymbol{0}, \boldsymbol{I}_n)$. *For any orthogonal* $(n \times n)$ *matrix* $\boldsymbol{Q}$ *and any continuous function* $f$, *we have* $\mathbb{E}_{\boldsymbol{z}}[f(\boldsymbol{z})] = \mathbb{E}_{\boldsymbol{z}}[f(\boldsymbol{Q}\boldsymbol{z})]$.

**Lemma 6.** *If* $x \sim \chi^2(n)$, *then we have:*

$$\mathbb{E}_x[x] = n, \quad \mathrm{Var}_x[x] = 2n.$$

**Lemma 7.** *(Nesterov & Spokoiny, 2017) Let* $f \in C_{L_2}^{2,2}(\mathbb{R}^n)$. *For all* $\boldsymbol{x}, \boldsymbol{y} \in \mathbb{R}^n$, *we have:*

$$|f(\boldsymbol{y}) - f(\boldsymbol{x}) - \langle \nabla f(\boldsymbol{x}), \boldsymbol{y} - \boldsymbol{x} \rangle - \frac{1}{2}\langle \nabla^2 f(\boldsymbol{x})(\boldsymbol{y} - \boldsymbol{x}), \boldsymbol{y} - \boldsymbol{x} \rangle| \leq \frac{L_2}{6}\|\boldsymbol{y} - \boldsymbol{x}\|^3.$$

**Lemma 8.** *(Nesterov & Spokoiny, 2017) Let* $\boldsymbol{z} \sim \mathcal{N}(\boldsymbol{0}, \boldsymbol{I}_n)$. *For* $0 \leq t \leq 2$, *we have:*

$$\mathbb{E}_{\boldsymbol{z}}[\|\boldsymbol{z}\|^t] \leq n^{t/2}.$$

*For* $t \geq 2$, *we have:*

$$n^{t/2} \leq \mathbb{E}_{\boldsymbol{z}}[\|\boldsymbol{z}\|^t] \leq (n + t)^{t/2}.$$

**Lemma 9.** *Let* $\boldsymbol{z} \sim \mathcal{N}(\boldsymbol{0}, \boldsymbol{I}_n)$. *For all* $\boldsymbol{y} \in \mathbb{R}^n$, *we have:*

$$\mathbb{E}_{\boldsymbol{z}}[\|\langle \boldsymbol{y}, \boldsymbol{z} \rangle \boldsymbol{z}\|^2] = (n + 2)\|\boldsymbol{y}\|^2.$$

*Proof.* Note that for any orthogonal $(n \times n)$-matrix $\boldsymbol{Q}$, we have:

$$\|\langle \boldsymbol{y}, \boldsymbol{Q}\boldsymbol{z} \rangle \boldsymbol{Q}\boldsymbol{z}\|^2 = \|\langle \boldsymbol{Q}^\mathsf{T} \boldsymbol{y}, \boldsymbol{z} \rangle \boldsymbol{z}\|^2, \quad \|\boldsymbol{Q}^\mathsf{T} \boldsymbol{y}\| = \|\boldsymbol{y}\|.$$

In accordance with Lemma 5, we can set $\boldsymbol{y} = [1, 0, \ldots, 0]^\mathsf{T}$, and only need to prove $\mathbb{E}_{\boldsymbol{z}}[\|\langle \boldsymbol{y}, \boldsymbol{z} \rangle \boldsymbol{z}\|^2] = n + 2$. Equipped with Lemma 6, we get:

$$\mathbb{E}_{\boldsymbol{z}}[\|\langle \boldsymbol{y}, \boldsymbol{z} \rangle \boldsymbol{z}\|^2] = \mathbb{E}_{\boldsymbol{z}}\left[\sum_{i=1}^n z_1^2 z_i^2\right] = \sum_{i=1}^n \mathbb{E}_{\boldsymbol{z}}[z_1^2 z_i^2] = \mathbb{E}_{\boldsymbol{z}_1}[z_1^4] + \mathbb{E}_{\boldsymbol{z}_1}[z_1^2]\sum_{i=2}^n \mathbb{E}_{\boldsymbol{z}}[z_i^2] = n + 2.$$

The proof is completed. $\qquad\square$

Here we provide the proof of Lemma 2.

*Proof.* **a)** The conclusion is clearly supported by Lemma 3 and Lemma 4.

**b)** Let $a_{\boldsymbol{z}}(\tau) = f(\boldsymbol{x} + \tau \boldsymbol{z}) - f(\boldsymbol{x}) - \tau \langle \nabla f(\boldsymbol{x}), \boldsymbol{z} \rangle - \frac{\tau^2}{2}\langle \nabla^2 f(\boldsymbol{x})\boldsymbol{z}, \boldsymbol{z} \rangle$. Lemma 7 implies that:

$$|a_{\boldsymbol{z}}(\pm\varepsilon)| \leq \frac{\varepsilon^3}{6} L_2 \|\boldsymbol{z}\|^3.$$

Note that:

$$\mathbb{E}_{\boldsymbol{z}}[\widehat{\nabla}f(\boldsymbol{\theta})] - \boldsymbol{P}\boldsymbol{P}^\mathsf{T}\nabla f(\boldsymbol{x})$$
$$= \frac{\boldsymbol{P}}{2\kappa\varepsilon} \int_{\mathbb{R}^q} [f(\boldsymbol{x}+\varepsilon\boldsymbol{P}\boldsymbol{z}) - f(\boldsymbol{x}-\varepsilon\boldsymbol{P}\boldsymbol{z}) - 2\varepsilon\langle\nabla f(\boldsymbol{z}), \boldsymbol{P}\boldsymbol{z}\rangle]\boldsymbol{z}e^{-\frac{1}{2}\|\boldsymbol{z}\|^2}d\boldsymbol{z}.$$

Therefore, in accordance with Lemma 8, we have:

$$\|\mathbb{E}_{\boldsymbol{z}}[\widehat{\nabla}f(\boldsymbol{\theta})] - \boldsymbol{P}\boldsymbol{P}^\mathsf{T}\nabla f(\boldsymbol{x})\|$$
$$\leq \frac{1}{2\kappa\varepsilon} \int_{\mathbb{R}^q} |f(\boldsymbol{x}+\varepsilon\boldsymbol{P}\boldsymbol{z}) - f(\boldsymbol{x}-\varepsilon\boldsymbol{P}\boldsymbol{z}) - 2\varepsilon\langle\nabla f(\boldsymbol{z}), \boldsymbol{P}\boldsymbol{z}\rangle|\|\boldsymbol{z}\|e^{-\frac{1}{2}\|\boldsymbol{z}\|^2}d\boldsymbol{z}$$
$$= \frac{1}{2\kappa\varepsilon} \int_{\mathbb{R}^q} |a_{\boldsymbol{P}\boldsymbol{z}}(\varepsilon) - a_{\boldsymbol{P}\boldsymbol{z}}(-\varepsilon)|\|\boldsymbol{z}\|e^{-\frac{1}{2}\|\boldsymbol{z}\|^2}d\boldsymbol{z}$$
$$\leq \frac{\varepsilon^2 L_2}{6\kappa} \int_{\mathbb{R}^q} \|\boldsymbol{z}\|^4 e^{-\frac{1}{2}\|\boldsymbol{z}\|^2}d\boldsymbol{z} \leq \frac{\varepsilon^2}{6} L_2(q+4)^2.$$

The proof is completed. □

**Theorem 2.** *Let $f(\boldsymbol{x}) = \boldsymbol{x}^\mathsf{T}\boldsymbol{H}\boldsymbol{x}$ and $\boldsymbol{z} \sim \mathcal{N}(\boldsymbol{0}, \boldsymbol{I}_q)$, where $\boldsymbol{H} \in \mathbb{R}^{d\times d}$ is positive definite. We have:*

$$\mathbb{E}_{\boldsymbol{z}}[\widehat{\nabla}f(\boldsymbol{\theta})] = \boldsymbol{P}\boldsymbol{P}^\mathsf{T}\nabla f(\boldsymbol{x}), \tag{6}$$
$$\mathbb{E}_{\boldsymbol{z}}[\|\widehat{\nabla}f(\boldsymbol{\theta})\|^2] = (q+2)\|\boldsymbol{P}^\mathsf{T}\nabla f(\boldsymbol{x})\|^2, \tag{7}$$
$$\mathbb{E}_{\boldsymbol{z}}\left[\frac{\langle\nabla f(\boldsymbol{x}), \widehat{\nabla}f(\boldsymbol{\theta})\rangle^2}{\|\boldsymbol{P}^\mathsf{T}\nabla f(\boldsymbol{x})\|^2\|\widehat{\nabla}f(\boldsymbol{\theta})\|^2}\right] = \frac{1}{q}. \tag{8}$$

*Proof.* It is straightforward to verify that

$$\widehat{\nabla}f(\boldsymbol{\theta}) = \boldsymbol{P}\langle\boldsymbol{P}^\mathsf{T}\nabla f(\boldsymbol{x}), \boldsymbol{z}\rangle\boldsymbol{z}.$$

Therefore, we can express the expected value as

$$\mathbb{E}_{\boldsymbol{z}}[\widehat{\nabla}f(\boldsymbol{\theta})] = \boldsymbol{P}\boldsymbol{P}^\mathsf{T}\nabla f(\boldsymbol{x}).$$

Combining this with Lemma 9, we find that

$$\mathbb{E}_{\boldsymbol{z}}[\|\widehat{\nabla}f(\boldsymbol{\theta})\|^2] = (q+2)\|\boldsymbol{P}^\mathsf{T}\nabla f(\boldsymbol{x})\|^2.$$

Additionally, it is important to note that for any orthogonal $(q \times q)$ matrix $\boldsymbol{Q}$, we have:

$$\mathbb{E}_{\boldsymbol{z}}\left[\frac{\langle\nabla f(\boldsymbol{x}), \widehat{\nabla}f(\boldsymbol{\theta})\rangle^2}{\|\boldsymbol{P}^\mathsf{T}\nabla f(\boldsymbol{x})\|^2\|\widehat{\nabla}f(\boldsymbol{\theta})\|^2}\right] = \mathbb{E}_{\boldsymbol{z}}\left[\frac{\langle\boldsymbol{P}^\mathsf{T}\nabla f(\boldsymbol{x}), \boldsymbol{z}\rangle^2}{\|\boldsymbol{P}^\mathsf{T}\nabla f(\boldsymbol{x})\|^2\|\boldsymbol{z}\|^2}\right]$$
$$= \mathbb{E}_{\boldsymbol{z}}\left[\frac{\langle\boldsymbol{P}^\mathsf{T}\nabla f(\boldsymbol{x}), \boldsymbol{Q}\boldsymbol{z}\rangle^2}{\|\boldsymbol{P}^\mathsf{T}\nabla f(\boldsymbol{x})\|^2\|\boldsymbol{Q}\boldsymbol{z}\|^2}\right]$$
$$= \mathbb{E}_{\boldsymbol{z}}\left[\frac{\langle\boldsymbol{Q}^\mathsf{T}\boldsymbol{P}^\mathsf{T}\nabla f(\boldsymbol{x}), \boldsymbol{z}\rangle^2}{\|\boldsymbol{Q}^\mathsf{T}\boldsymbol{P}^\mathsf{T}\nabla f(\boldsymbol{x})\|^2\|\boldsymbol{z}\|^2}\right].$$

In accordance with Lemma 5, we can set $\boldsymbol{P}^\mathsf{T}\nabla f(\boldsymbol{x}) = [1, 0, \ldots, 0]^\mathsf{T}$. Thus, we have:

$$\mathbb{E}_{\boldsymbol{z}}\left[\frac{\langle\nabla f(\boldsymbol{x}), \widehat{\nabla}f(\boldsymbol{\theta})\rangle^2}{\|\boldsymbol{P}^\mathsf{T}\nabla f(\boldsymbol{x})\|^2\|\widehat{\nabla}f(\boldsymbol{\theta})\|^2}\right] = \mathbb{E}_{\boldsymbol{z}}\left[\frac{z_1^2}{\|\boldsymbol{z}\|^2}\right] = \frac{1}{q}.$$

The proof is completed. □

To demonstrate the convergence of SVD-0 with SGD, we can structure our analysis into two main segments. The first segment examines the convergence behavior of the SVD-0 solution process while keeping the projection matrix $\boldsymbol{P}$ constant. The second segment evaluates the impact of performing lazy updates to $\boldsymbol{P}$. Through these assessments, we aim to establish the global convergence of SVD-0, specifically in the context of a single layer.

In the initial stage, while $\boldsymbol{P}$ remains constant, we can reformulate the original SVD-0 problem as an optimization problem constrained within that subspace. We define $h(\boldsymbol{y}) = f(\boldsymbol{x} + \boldsymbol{P}\boldsymbol{y})$, $h_\varepsilon(\boldsymbol{y}) = \mathbb{E}_{\boldsymbol{z}}[h(\boldsymbol{y} + \varepsilon\boldsymbol{z})]$, and $g_\varepsilon(\boldsymbol{y}) = \frac{h(\boldsymbol{y}+\varepsilon\boldsymbol{z})-f(\boldsymbol{y})}{\varepsilon}\boldsymbol{z}$. According to Lemma 10, if $f$ demonstrates first $L_1$-smoothness, then $h$ will also exhibit first $L_1$-smoothness.

**Lemma 10.** *Let $h(\boldsymbol{y}) = f(\boldsymbol{x} + \boldsymbol{P}\boldsymbol{y})$, where $f \in C_{L_1}^{1,1}(\mathbb{R}^d)$, and $\boldsymbol{P}^\mathsf{T}\boldsymbol{P} = \boldsymbol{I}$, then we have $h \in C_{L_1}^{1,1}(\mathbb{R}^q)$.*

*Proof.* The following demonstrates that if $f$ is first $L_1$-smooth, then $h$ is also first $L_1$-smooth. For any $\boldsymbol{y}_1 \in \mathbb{R}^q$ and $\boldsymbol{y}_2 \in \mathbb{R}^q$, we have:

$$
\begin{aligned}
\|\nabla h(\boldsymbol{y}_1) - \nabla h(\boldsymbol{y}_2)\| &= \left\|\boldsymbol{P}^\mathsf{T}\nabla(f(\boldsymbol{x} + \boldsymbol{P}\boldsymbol{y}_1) - \boldsymbol{P}^\mathsf{T}\nabla(f(\boldsymbol{x} + \boldsymbol{P}\boldsymbol{y}_2)\right\| \\
&\leq \left\|\boldsymbol{P}^\mathsf{T}\right\|\left\|\nabla(f(\boldsymbol{x} + \boldsymbol{P}\boldsymbol{y}_1) - \nabla(f(\boldsymbol{x} + \boldsymbol{P}\boldsymbol{y}_2)\right\| \\
&\leq L_1\left\|\boldsymbol{P}(\boldsymbol{y}_1 - \boldsymbol{y}_2)\right\| \\
&= L_1\left\|\boldsymbol{y}_1 - \boldsymbol{y}_2\right\|.
\end{aligned}
$$

The proof is completed. $\qquad\square$

We then analyze the convergence of SVD-0 while maintaining a fixed subspace.

**Lemma 11.** *(Nesterov & Spokoiny, 2017) Let $f \in C_{L_1}^{1,1}(\mathbb{R})$. Then, for any $\boldsymbol{x} \in \mathbb{R}$, we have:*

$$
E_{\boldsymbol{z}}[\|g_\varepsilon(\boldsymbol{x})\|^2] = E_{\boldsymbol{z}}\left[\|\frac{f(\boldsymbol{x} + \varepsilon\boldsymbol{z}) - f(\boldsymbol{x})}{\varepsilon}\|^2\right] \leq 4(n+4)\|\nabla f_\varepsilon(\boldsymbol{x})\|^2 + 3\varepsilon^2 L_1^2(f)(n+4)^3, \quad (9)
$$

*and*

$$
\|\nabla f(\boldsymbol{x})\|^2 \leq 2\|\nabla f_\varepsilon(\boldsymbol{x})\|^2 + \frac{\varepsilon^2}{2}L_1^2(f)(n+6)^3, \quad (10)
$$

*where $f_\varepsilon(\boldsymbol{x}) = \mathbb{E}_{\boldsymbol{z}}[f(\boldsymbol{x} + \varepsilon\boldsymbol{z})]$.*

**Lemma 12.** *Let $\boldsymbol{y}^* = \arg\min_{\boldsymbol{x} \in \mathbb{R}^q} h(\boldsymbol{y})$, where $h \in C_{L_1}^{1,1}(\mathbb{R}^q)$ and $h$ is non-convex. Suppose $\mathcal{E}_k = (\boldsymbol{z}_0, \boldsymbol{z}_1, \cdots, \boldsymbol{z}_{k-1}, \boldsymbol{z}_k)$, where $\boldsymbol{z}_k \sim \mathcal{N}(0, \boldsymbol{I}_q)$ and $\eta = \frac{1}{4(q+4)L_1}$. $\{\boldsymbol{y}_k\}_{k>0}$ is the sequence generated by Algorithm 3. Let $\phi_0 = h(\boldsymbol{y}_0)$, and for $k \geq 1$, $\phi_k = \mathbb{E}_{\mathcal{E}_{k-1}}[h(\boldsymbol{y}_k)]$. For the $\boldsymbol{P}$ defined in Proposition 1, which is fixed, we have:*

$$
\phi_{k+1} - \phi_k \leq -\frac{1}{4}\eta\mathbb{E}_{\mathcal{E}_k}\left[\|\nabla h(\boldsymbol{y}_k)\|^2\right] + \frac{\varepsilon^2(q+6)^3}{8}L_1^2 + \frac{3\varepsilon^2(q+4)}{32}L_1 \quad (11)
$$

*Proof.* If we have a fixed subspace $\boldsymbol{P} \in \mathbb{R}^{d \times q}$, we can reformulate the optimization objective as follows:

$$
\min_{\boldsymbol{y} \in \mathbb{R}^q} h(\boldsymbol{y}) := f(\boldsymbol{x} + \boldsymbol{P}\boldsymbol{y}),
$$

Let $\boldsymbol{y}_0$ represent the initial point, and let $\{\eta_k\}_{k \geq 0}$ be a sequence of positive real numbers. We will consider the randomized gradient search algorithm $\mathcal{RG}_\varepsilon(\varepsilon > 0)$:

1. Generate $\boldsymbol{z}_k$ and the corresponding $g_\varepsilon(\boldsymbol{y}_k)$, where $\boldsymbol{z}_k \sim \mathcal{N}(\boldsymbol{0}, \boldsymbol{I}_q)$.

2. Update $\boldsymbol{y}_{k+1} = \boldsymbol{y}_k - \eta_k g_\varepsilon(\boldsymbol{y}_k)$.

Our goal is to estimate the evolution of the function $h_\varepsilon$ after one iteration of this algorithm.

Given that $h$ is $L_1$-Lipschitz continuous for its first derivative, and $h_\varepsilon$ is $L_\varepsilon$-Lipschitz continuous for its first derivative (where $L_\varepsilon \leq L_1$)(Nesterov & Spokoiny, 2017), we have:

$$
h_\varepsilon(\boldsymbol{y}_{k+1}) \leq h_\varepsilon(\boldsymbol{y}_k) - \eta_k\langle\nabla h_\varepsilon(\boldsymbol{y}_k), g_\varepsilon(\boldsymbol{y}_k)\rangle + \frac{1}{2}\eta_k^2 L_\varepsilon\|g_\varepsilon(\boldsymbol{y}_k)\|^2.
$$

Taking expectation with respect to $\boldsymbol{z}_k$, we have:

$$
\mathbb{E}_{\boldsymbol{z}_k}[h_\varepsilon(\boldsymbol{y}_{k+1})] \leq h_\varepsilon(\boldsymbol{y}_k) - \eta_k\|\nabla h_\varepsilon(\boldsymbol{y}_k)\|^2 + \frac{1}{2}\eta_k^2 L_\varepsilon\,\mathbb{E}_{\boldsymbol{z}_k}[\|g_\varepsilon(\boldsymbol{y}_k)\|^2].
$$

Since $h \in C^{1,1}(\mathbb{R}^q)$, from Lemma 11, we have:

$$\mathbb{E}_{\boldsymbol{z}_k}[h_\varepsilon(\boldsymbol{y}_{k+1})] \leq h_\varepsilon(\boldsymbol{y}_k) - \eta_k \|\nabla h_\varepsilon(\boldsymbol{y}_k)\|^2$$
$$+ \frac{1}{2}\eta_k^2 L_1 \left(4(q+4)\|\nabla h_\varepsilon(\boldsymbol{y}_k)\|^2 + 3\varepsilon^2 L_1^2(q+4)^3\right).$$

Setting $\eta_k = \hat{\eta} = \frac{1}{4(q+4)L_1}$, we get:

$$\mathbb{E}_{\boldsymbol{z}_k}[h_\varepsilon(\boldsymbol{y}_{k+1})] \leq h_\varepsilon(\boldsymbol{y}_k) - \frac{1}{2}\hat{\eta}\|\nabla h_\varepsilon(\boldsymbol{y}_k)\|^2 + \frac{3\varepsilon^2}{32}L_1(q+4).$$

Taking the expectation with respect to $\mathcal{E}_k$, we get:

$$\phi_{k+1} \leq \phi_k - \frac{1}{2}\hat{\eta}\mathbb{E}_{\mathcal{E}_k}[\|\nabla h_\varepsilon(\boldsymbol{y}_k)\|^2] + \frac{3\varepsilon^2(q+4)}{32}L_1,$$

From Lemma 11, we have $\mathbb{E}_{\mathcal{E}_k}[\|\nabla h(\boldsymbol{y}_k)\|^2] \leq 2\mathbb{E}_{\mathcal{E}_k}[\|\nabla h_\varepsilon(\boldsymbol{y}_k)\|^2] + \frac{\varepsilon^2(q+6)^3}{2}L_1^2$. Therefore:

$$\phi_{k+1} - \phi_k \leq -\frac{1}{4}\hat{\eta}\mathbb{E}_{\mathcal{E}_k}\left[\|\nabla h(\boldsymbol{y}_k)\|^2\right] + \frac{\varepsilon^2(q+6)^3}{8}L_1^2 + \frac{3\varepsilon^2(q+4)}{32}L_1. \qquad (12)$$

The proof is completed. $\qquad\square$

Next, we need to assess the randomness of our random subspace. According to Lemma 16, if we obtain the projection matrix using Algorithm 1, the expected value can be expressed as $\mathbb{E}[\boldsymbol{P}\boldsymbol{P}^T] = \frac{q}{d}\boldsymbol{I}$. In this equation, $q$ represents the dimension of the subspace, $d$ indicates the dimension of the original space, and $\boldsymbol{P}$ is defined as $\boldsymbol{V} \otimes \boldsymbol{U}$.

**Lemma 13.** *Let matrix $\boldsymbol{A} = (\boldsymbol{a}_1, \boldsymbol{a}_2, \cdots, \boldsymbol{a}_r) \in \mathbb{R}^{n \times r}$ be composed of column vectors $\boldsymbol{a}_k$ which are mutually independent and $\boldsymbol{a}_k \in \mathcal{N}(0, \boldsymbol{I}_n)$. Suppose Gram-Schmidt process $\boldsymbol{u}_k = \boldsymbol{a}_k - \sum_{s=1}^{k-1} \langle \boldsymbol{a}_k, \boldsymbol{e}_s \rangle \boldsymbol{e}_s$ and $\boldsymbol{e}_k = \frac{\boldsymbol{u}_k}{\|\boldsymbol{u}_k\|}$. $[\boldsymbol{a}_k]_i \leftrightarrow [\boldsymbol{a}_k]_j$ represents the exchange of the $i$-th element and the $j$-th element of $\boldsymbol{a}_k$, while all other elements remain unchanged. $[\boldsymbol{a}_k]_i = -1 \times [\boldsymbol{a}_k]_i$ signifies that only the $i$-th element of $\boldsymbol{a}_k$ is multiplied by $-1$, while all other elements remain unchanged. Suppose $f(\boldsymbol{A}, \boldsymbol{U}, \boldsymbol{E})$ be a function of the matrix $\boldsymbol{A}$, $\boldsymbol{U} = (\boldsymbol{u}_1, \boldsymbol{u}_2, \cdots, \boldsymbol{u}_r)$ and $\boldsymbol{E} = (\boldsymbol{e}_1, \boldsymbol{e}_2, \cdots, \boldsymbol{e}_r)$, then*

*(1) if $[\boldsymbol{a}_k]_i \leftrightarrow [\boldsymbol{a}_k]_j$ or $[\boldsymbol{a}_k]_i = -1 \times [\boldsymbol{a}_k]_i$, $\mathbb{E}[f]$ remain unchanged;*

*(2) if $[\boldsymbol{a}_k]_i \leftrightarrow [\boldsymbol{a}_k]_j \Rightarrow [\boldsymbol{u}_k]_i \leftrightarrow [\boldsymbol{u}_k]_j$ and $[\boldsymbol{e}_k]_i \leftrightarrow [\boldsymbol{e}_k]_j$;*

*(3) if $[\boldsymbol{a}_k]_i = -1 \times [\boldsymbol{a}_k]_i \Rightarrow [\boldsymbol{u}_k]_i = -1 \times [\boldsymbol{u}_k]_i$, $[\boldsymbol{e}_k]_i = -1 \times [\boldsymbol{e}_k]_i$, $[\boldsymbol{u}_k]_j = 1 \times [\boldsymbol{u}_k]_j$, and $[\boldsymbol{e}_k]_j = 1 \times [\boldsymbol{e}_k]_j$, where $i \neq j$;*

*(4) $\mathbb{E}\left[\frac{[\boldsymbol{u}_k]_i^2}{\langle \boldsymbol{u}_k, \boldsymbol{u}_k \rangle}\right] = \frac{1}{n}$;*

*(5) $\mathbb{E}\left[\frac{[\boldsymbol{u}_k]_i[\boldsymbol{u}_k]_j}{\langle \boldsymbol{u}_k, \boldsymbol{u}_k \rangle}\right] = 0$, where $i \neq j$.*

*Proof.* In real analysis, a matrix $\boldsymbol{A}$ usually possesses full rank when it follows a Gaussian distribution, and it is common for both $\boldsymbol{u}_k$ and $\boldsymbol{e}_k$ to be non-zero.

(1) Given that $\boldsymbol{a}_k$ is independently and identically distributed, this condition clearly applies.

(2) For the base case $k = 1$, it is obviously true. Assume that the result holds for all $k = 1, 2, \cdots, k-1$, where $k \geq 2$, then $[\boldsymbol{a}_k]_i \leftrightarrow [\boldsymbol{a}_k]_j \Rightarrow [\boldsymbol{u}_k]_i = [\boldsymbol{a}_k]_i - \sum_{s=1}^{k-1} \langle \boldsymbol{a}_k, \boldsymbol{e}_s \rangle [\boldsymbol{e}_s]_i$, $[\boldsymbol{u}_k]_j = [\boldsymbol{a}_k]_j - \sum_{s=1}^{k-1} \langle \boldsymbol{a}_k, \boldsymbol{e}_s \rangle [\boldsymbol{e}_s]_j$, $[\boldsymbol{e}_k]_i = \frac{[\boldsymbol{u}_k]_i}{\|\boldsymbol{u}_k\|}$, and $[\boldsymbol{e}_k]_j = \frac{[\boldsymbol{u}_k]_j}{\|\boldsymbol{u}_k\|}$.

Thus, by strong induction, we have $[\boldsymbol{u}_k]_i \leftrightarrow [\boldsymbol{u}_k]_j$ and $[\boldsymbol{e}_k]_i \leftrightarrow [\boldsymbol{e}_k]_j$.

(3) For base case $k = 1$, it obviously holds. Assume the result holds for all $k = 1, 2, \cdots, k-1$, where $k \geq 2$, then

$$[\boldsymbol{a}_k]_i = -1 \times [\boldsymbol{a}_k]_i \Rightarrow \begin{cases} [\boldsymbol{u}_k]_i = [\boldsymbol{a}_k]_i \times (-1) - \sum_{s=1}^{k-1} \langle \boldsymbol{a}_k, \boldsymbol{e}_s \rangle [\boldsymbol{e}_s]_i \times (-1) = [\boldsymbol{u}_k]_i \times (-1) \\ [\boldsymbol{u}_k]_j = [\boldsymbol{u}_k]_j \times 1, i \neq j \end{cases}$$

$$\Rightarrow \begin{cases} [\boldsymbol{e}_k]_i \times (-1) = \frac{[\boldsymbol{u}_k]_i}{\|\boldsymbol{u}_k\|} \times (-1) \\ [\boldsymbol{e}_k]_j = [\boldsymbol{e}_k]_j \times 1, j \neq i \end{cases}$$

By strong induction, we have $[\boldsymbol{u}_k]_i = -1 \times [\boldsymbol{u}_k]_i$, $[\boldsymbol{e}_k]_i = -1 \times [\boldsymbol{e}_k]_i$, $[\boldsymbol{u}_k]_j = 1 \times [\boldsymbol{u}_k]_j$, and $[\boldsymbol{e}_k]_j = 1 \times [\boldsymbol{e}_k]_j$, where $i \neq j$. $\qquad \square$

(4) Since $\left| \frac{[\boldsymbol{u}_k]_i^2}{\langle \boldsymbol{u}_k, \boldsymbol{u}_k \rangle} \right| \leq 1$, $\mathbb{E}\left[ \frac{[\boldsymbol{u}_k]_i^2}{\langle \boldsymbol{u}_k, \boldsymbol{u}_k \rangle} \right]$ exists. $[\boldsymbol{a}_k]_i \leftrightarrow [\boldsymbol{a}_k]_j \Rightarrow \frac{[\boldsymbol{u}_k]_i^2}{\langle \boldsymbol{u}_k, \boldsymbol{u}_k \rangle} \leftrightarrow \frac{[\boldsymbol{u}_k]_j^2}{\langle \boldsymbol{u}_k, \boldsymbol{u}_k \rangle}$.

Thus, $\mathbb{E}\left[ \frac{[\boldsymbol{u}_k]_i^2}{\langle \boldsymbol{u}_k, \boldsymbol{u}_k \rangle} \right] \times n = \sum_{s=1}^n \mathbb{E}\left[ \frac{[\boldsymbol{u}_k]_s^2}{\langle \boldsymbol{u}_k, \boldsymbol{u}_k \rangle} \right] = \mathbb{E}\left[ \frac{\langle \boldsymbol{u}_k, \boldsymbol{u}_k \rangle}{\langle \boldsymbol{u}_k, \boldsymbol{u}_k \rangle} \right] = 1 \Rightarrow \mathbb{E}\left[ \frac{[\boldsymbol{u}_k]_i^2}{\langle \boldsymbol{u}_k, \boldsymbol{u}_k \rangle} \right] = \frac{1}{n}$.

(5) Since $\left| \frac{[\boldsymbol{u}_k]_i [\boldsymbol{u}_k]_j}{\langle \boldsymbol{u}_k, \boldsymbol{u}_k \rangle} \right| \leq \left| \frac{[\boldsymbol{u}_k]_i^2 + [\boldsymbol{u}_k]_j^2}{2\langle \boldsymbol{u}_k, \boldsymbol{u}_k \rangle} \right| \leq 1$, $\mathbb{E}\left[ \frac{[\boldsymbol{u}_k]_i [\boldsymbol{u}_k]_j}{\langle \boldsymbol{u}_k, \boldsymbol{u}_k \rangle} \right]$ exists.

$$[\boldsymbol{a}_k]_i = [\boldsymbol{a}_k]_i \times -1 \Rightarrow \mathbb{E}\left[ \frac{[\boldsymbol{u}_k]_i [\boldsymbol{u}_k]_j}{\langle \boldsymbol{u}_k, \boldsymbol{u}_k \rangle} \right] = \mathbb{E}\left[ \frac{-[\boldsymbol{u}_k]_i [\boldsymbol{u}_k]_j}{\langle \boldsymbol{u}_k, \boldsymbol{u}_k \rangle} \right] = 0, \text{ where } i \neq j.$$

**Lemma 14.** *Let $\boldsymbol{A} \in \mathbb{R}^{n \times r}$ be a matrix with independent standard normal entries, i.e., each element of $\boldsymbol{A}$ is an i.i.d. $\mathcal{N}(0,1)$ random variable. Suppose $\boldsymbol{A}$ undergoes QR decomposition via the Gram-Schmidt process to yield a column-orthogonal matrix $\boldsymbol{Q} \in \mathbb{R}^{n \times r}$ with orthonormal columns $\boldsymbol{e}_1, \boldsymbol{e}_2, \ldots, \boldsymbol{e}_r$ and an upper triangular matrix $\boldsymbol{R} \in \mathbb{R}^{r \times r}$. Then, for each $k = 1, 2, \ldots, r$, the expected value of the outer product of the k-th orthonormal column vector $\boldsymbol{e}_k$ of $\boldsymbol{Q}$ is given by:*

$$\mathbb{E}[\boldsymbol{e}_k \boldsymbol{e}_k^T] = \frac{1}{n} \boldsymbol{I},$$

*where $\boldsymbol{I}$ is the $n \times n$ identity matrix.*

*Proof.* By the Gram-Schmidt process, we have $\boldsymbol{e}_k = \frac{\boldsymbol{u}_k}{\|\boldsymbol{u}_k\|}$, where $\boldsymbol{u}_k = \boldsymbol{a}_k - \sum_{s=1}^{k-1} \langle \boldsymbol{a}_k, \boldsymbol{e}_s \rangle \boldsymbol{e}_s$. Thus, $\boldsymbol{e}_k \boldsymbol{e}_k^T = \frac{\boldsymbol{u}_k \boldsymbol{u}_k^T}{\langle \boldsymbol{u}_k, \boldsymbol{u}_k \rangle}$.

The $(i, j)$-th entry of $\mathbb{E}[\boldsymbol{e}_k \boldsymbol{e}_k^T]$ can be written as:

$$\mathbb{E}[[\boldsymbol{e}_k \boldsymbol{e}_k^T]_{ij}] = \mathbb{E}\left[ \frac{[\boldsymbol{u}_k]_i [\boldsymbol{u}_k]_j}{\langle \boldsymbol{u}_k, \boldsymbol{u}_k \rangle} \right].$$

For diagonal entries ($i = j$): When $i = j$, from Lemma 13(4), we have:

$$\mathbb{E}[[\boldsymbol{e}_k \boldsymbol{e}_k^T]_{ii}] = \mathbb{E}\left[ \frac{[\boldsymbol{u}_k]_i^2}{\langle \boldsymbol{u}_k, \boldsymbol{u}_k \rangle} \right] = \frac{1}{n}.$$

For off-diagonal entries ($i \neq j$): When $i \neq j$, from Lemma 13(5), we have:

$$\mathbb{E}[[\boldsymbol{e}_k \boldsymbol{e}_k^T]_{ij}] = \mathbb{E}\left[ \frac{[\boldsymbol{u}_k]_i [\boldsymbol{u}_k]_j}{\langle \boldsymbol{u}_k, \boldsymbol{u}_k \rangle} \right] = 0.$$

Combining these two cases, we conclude that $\mathbb{E}[\boldsymbol{e}_k \boldsymbol{e}_k^T]$ is a diagonal matrix with all diagonal entries equal to $\frac{1}{n}$. Thus,

$$\mathbb{E}[\boldsymbol{e}_k \boldsymbol{e}_k^T] = \frac{1}{n} \boldsymbol{I},$$

where $\boldsymbol{I}$ is the $n \times n$ identity matrix. The proof is completed. $\qquad \square$

**Lemma 15.** *Let $\boldsymbol{A} \in \mathbb{R}^{n \times r}$ be a matrix with independent standard normal entries, i.e., each element of $\boldsymbol{A}$ is an i.i.d. $\mathcal{N}(0,1)$ random variable. Suppose $\boldsymbol{A}$ undergoes QR decomposition to yield an orthogonal matrix $\boldsymbol{Q} \in \mathbb{R}^{n \times r}$ with orthonormal columns and an upper triangular matrix $\boldsymbol{R} \in \mathbb{R}^{r \times r}$. Then, the expected value of the outer product of the matrix $\boldsymbol{Q}$ with itself is given by:*

$$\mathbb{E}[\boldsymbol{Q}\boldsymbol{Q}^T] = \frac{r}{n} \boldsymbol{I}$$

*where $\boldsymbol{I}$ is the $n \times n$ identity matrix.*

*Proof.* The QR decomposition of the matrix $\boldsymbol{A}$ is expressed as $\boldsymbol{A} = \boldsymbol{Q}\boldsymbol{R}$, where $\boldsymbol{Q}$ is an orthogonal matrix with columns denoted as $\boldsymbol{e}_1, \boldsymbol{e}_2, \ldots, \boldsymbol{e}_r$, and $\boldsymbol{R}$ is an upper triangular matrix. Since $\boldsymbol{Q}$ is orthogonal, it satisfies the condition $\boldsymbol{Q}\boldsymbol{Q}^T = \boldsymbol{I}_r$, where $\boldsymbol{I}_r$ represents the $r \times r$ identity matrix. Our objective is to compute $\mathbb{E}[\boldsymbol{Q}\boldsymbol{Q}^T]$. By leveraging the linearity of expectation and the fact that the columns of $\boldsymbol{Q}$ are orthonormal, we find:

$$\mathbb{E}[\boldsymbol{Q}\boldsymbol{Q}^T] = \mathbb{E}\left[\sum_{k=1}^{r} \boldsymbol{e}_k \boldsymbol{e}_k^T\right] = \sum_{k=1}^{r} \mathbb{E}[\boldsymbol{e}_k \boldsymbol{e}_k^T].$$

From Lemma 14, we know that $\mathbb{E}[\boldsymbol{e}_k \boldsymbol{e}_k^T] = \frac{1}{n}\boldsymbol{I}$ for each $k$. Therefore:

$$\mathbb{E}[\boldsymbol{Q}\boldsymbol{Q}^T] = \sum_{k=1}^{r} \frac{1}{n}\boldsymbol{I} = \frac{r}{n}\boldsymbol{I}.$$

The proof is completed. $\qquad\square$

**Lemma 16.** *Let $\boldsymbol{A}_1 \in \mathbb{R}^{m \times r}$ and $\boldsymbol{A}_2 \in \mathbb{R}^{n \times r}$ be matrices with independent standard normal entries, i.e., each element of $\boldsymbol{A}_1$ and $\boldsymbol{A}_2$ is an i.i.d. $\mathcal{N}(0,1)$ random variable. Suppose $\boldsymbol{A}_1$ and $\boldsymbol{A}_2$ undergo QR decomposition to yield orthogonal matrices $\boldsymbol{Q}_1 \in \mathbb{R}^{m \times r}$ and $\boldsymbol{Q}_2 \in \mathbb{R}^{n \times r}$ with orthonormal columns, respectively. Define $\boldsymbol{P} = \boldsymbol{Q}_2 \otimes \boldsymbol{Q}_1$, where $\otimes$ denotes the Kronecker product. Then, the expected value of the outer product of the matrix $\boldsymbol{P}$ with itself is given by:*

$$\mathbb{E}[\boldsymbol{P}\boldsymbol{P}^T] = \frac{r^2}{mn}\boldsymbol{I},$$

*where $\boldsymbol{I}$ is the $mn \times mn$ identity matrix.*

*Proof.* The Kronecker product $\boldsymbol{P} = \boldsymbol{Q}_2 \otimes \boldsymbol{Q}_1$ produces a matrix $\boldsymbol{P} \in \mathbb{R}^{mn \times r^2}$. According to Lemma 15, we know that $\boldsymbol{Q}_1\boldsymbol{Q}_1^T = \frac{r}{m}\boldsymbol{I}$ and $\boldsymbol{Q}_2\boldsymbol{Q}_2^T = \frac{r}{n}\boldsymbol{I}$. Our goal is to compute $\mathbb{E}[\boldsymbol{P}\boldsymbol{P}^T]$. By utilizing the properties of the Kronecker product, we can proceed with our computation:

$$\mathbb{E}[\boldsymbol{P}\boldsymbol{P}^T] = \mathbb{E}[(\boldsymbol{Q}_2 \otimes \boldsymbol{Q}_1)(\boldsymbol{Q}_2^T \otimes \boldsymbol{Q}_1^T)] = \mathbb{E}[(\boldsymbol{Q}_2\boldsymbol{Q}_2^T)] \otimes \mathbb{E}[(\boldsymbol{Q}_1\boldsymbol{Q}_1^T)] = \frac{r^2}{mn}\boldsymbol{I} \otimes \boldsymbol{I} = \frac{r^2}{mn}\boldsymbol{I}$$

The proof is completed. $\qquad\square$

Now we can assess the impact of the lazy updates to $\boldsymbol{P}$. Here we provide the proof of Theorem 1.

*Proof.* Let $\mathcal{P}_j = (\boldsymbol{P}_0, \boldsymbol{P}_1, \ldots, \boldsymbol{P}_j)$, where $\boldsymbol{P}_j$ is the sequence generated by Proposition 1 and $j \leq K$. According to Lemma 10 and Lemma 12, when the subspace is fixed, we can transform the original problem $f \in C_{L_1}^{1,1}(\mathbb{R}^d)$ into $h \in C_{L_1}^{1,1}(\mathbb{R}^q)$ using the transformation $h(\boldsymbol{y}) = f(\boldsymbol{x} + \boldsymbol{P}\boldsymbol{y})$. Consider the following update rule:

$$\boldsymbol{y}_{j,0} = 0, h_j(\boldsymbol{y}) = f(\boldsymbol{x}_{jF} + \boldsymbol{P}_j\boldsymbol{y}), \forall j \in 0, 1, \cdots, K-1 \tag{13}$$

$$\boldsymbol{y}_{j,k} = \boldsymbol{y}_{j,k-1} - \eta\widehat{\nabla}h_j(\boldsymbol{y}_{j,k-1}), \forall k \in 0, 1, \cdots, F \tag{14}$$

$$\boldsymbol{x}_{jF+k} = \boldsymbol{x}_{jF} + \boldsymbol{P}_j\boldsymbol{y}_k, \tag{15}$$

In the $j$-th subspace, the projection matrix $\boldsymbol{P}_j$ is constant, allowing us to accumulate the changes of $\phi$ within this subspace. By applying Lemma 12, we obtain:

$$\phi_{(j+1)F} - \phi_{jF} \leq -\frac{1}{4}\hat{\eta}\sum_{i=0}^{K-1}\mathbb{E}_{\mathcal{E}_{jF+i}}\left[\|\nabla h_j(\boldsymbol{y}_{j,i})\|^2\right] + \frac{\varepsilon^2(q+6)^3}{8}KL_1^2 + \frac{3\varepsilon^2(q+4)}{32}KL_1 \tag{16}$$

$$\leq -\frac{1}{4}\hat{\eta}\mathbb{E}_{\mathcal{E}_{jF}}\left[\|\nabla h_j(\boldsymbol{y}_{j,0})\|^2\right] + \frac{\varepsilon^2(q+6)^3}{8}KL_1^2 + \frac{3\varepsilon^2(q+4)}{32}KL_1. \tag{17}$$

Furthermore, we note that $\nabla h_j(\boldsymbol{y}_{j,0}) = (\boldsymbol{P}_j)^\mathsf{T}\nabla f(\boldsymbol{x}_{jF})$. By taking expectations over the overall historical projection matrix $\mathcal{P}_j$ and applying Lemma 16, we find that $\mathbb{E}[\boldsymbol{P}_j(\boldsymbol{P}_j)^\mathsf{T}] = \frac{q}{d}\boldsymbol{I}$, with $\boldsymbol{P}_j$

being independent of $\boldsymbol{x}_{jF}$. Thus, we obtain:

$$\mathbb{E}_{\mathcal{P}_{j+1}}[\phi_{(j+1)F}] - \mathbb{E}_{\mathcal{P}_j}[\phi_{jF}] \leq -\frac{1}{4}\hat{\eta}\mathbb{E}_{\mathcal{E}_{jF},\mathcal{P}_j}\left[\|(\boldsymbol{P}_j)^\mathsf{T}\nabla f(\boldsymbol{x}_{jF})\|^2\right] + \frac{\varepsilon^2(q+6)^3}{8}KL_1^2 + \frac{3\varepsilon^2(q+4)}{32}KL_1 \tag{18}$$

$$= -\frac{q}{4d}\hat{\eta}\mathbb{E}_{\mathcal{E}_{jF},\mathcal{P}_j}\left[\|\nabla f(\boldsymbol{x}_{jF})\|^2\right] + \frac{\varepsilon^2(q+6)^3}{8}KL_1^2 + \frac{3\varepsilon^2(q+4)}{32}KL_1. \tag{19}$$

Assuming $f(\boldsymbol{x}) \geq f^*$ holds for all $\boldsymbol{x} \in \mathbb{R}^d$, and letting $T = KF$, summing the inequality yields:

$$\mathbb{E}_{\mathcal{P}_{K-1}}[\phi_T] \leq \mathbb{E}_{\mathcal{P}_0}[\phi_0] - \frac{q}{4d}\hat{\eta}\sum_{j=0}^{K-1}\mathbb{E}_{\mathcal{E}_{jF},\mathcal{P}_j}\left[\|\nabla f(\boldsymbol{x}_{jF})\|^2\right] + T\frac{\varepsilon^2(q+6)^3}{8}L_1^2 + T\frac{3\varepsilon^2(q+4)}{32}L_1. \tag{20}$$

Since $\mathbb{E}_{\mathcal{P}_{K-1}}[\phi_T] \geq f^*$, we have:

$$f^* \leq \mathbb{E}_{\mathcal{P}_0}[\phi_0] - \frac{q}{4d}\hat{\eta}\sum_{j=0}^{K-1}\mathbb{E}_{\mathcal{E}_{jF},\mathcal{P}_j}\left[\|\nabla f(\boldsymbol{x}_{jF})\|^2\right] + T\frac{\varepsilon^2(q+6)^3}{8}L_1^2 + T\frac{3\varepsilon^2(q+4)}{32}L_1. \tag{21}$$

Rearranging the inequality, we get:

$$\frac{q}{4d}\hat{\eta}\sum_{j=0}^{K-1}\mathbb{E}_{\mathcal{E}_{jF},\mathcal{P}_j}\left[\|\nabla f(\boldsymbol{x}_{jF})\|^2\right] \leq \mathbb{E}_{\mathcal{P}_0}[\phi_0] - f^* + T\frac{\varepsilon^2(q+6)^3}{8}L_1^2 + T\frac{3\varepsilon^2(q+4)}{32}L_1. \tag{22}$$

Substituting $\hat{\eta} = \frac{1}{4(q+4)L_1}$, we obtain:

$$\frac{q}{16d(q+4)L_1}\sum_{j=0}^{K-1}\mathbb{E}_{\mathcal{E}_{jF},\mathcal{P}_j}\left[\|\nabla f(\boldsymbol{x}_{jF})\|^2\right] \leq \mathbb{E}_{\mathcal{P}_0}[\phi_0] - f^* + T\frac{\varepsilon^2(q+6)^3}{8}L_1^2 + T\frac{3\varepsilon^2(q+4)}{32}L_1. \tag{23}$$

Thus, we have:

$$\frac{1}{T}\sum_{k=0}^{T-1}\mathbb{E}_{\mathcal{E}_k,\mathcal{P}_{\lfloor k/F\rfloor}}\left[\|\nabla f(\boldsymbol{x}_k)\|^2\right] \leq \frac{16(q+4)dL_1(\mathbb{E}_{\mathcal{P}_0}[\phi_0] - f^*)}{qT} + \frac{2\varepsilon^2(q+6)^3(q+4)d}{q}L_1^3 + \frac{3\varepsilon^2(q+4)^2 d}{2q}L_1^2. \tag{24}$$

To ensure $\frac{1}{T}\sum_{k=0}^{T-1}\mathbb{E}_{\mathcal{E}_k,\mathcal{P}_{\lfloor k/F\rfloor}}\left[\|\nabla f(\boldsymbol{x}_k)\|^2\right] \leq \epsilon$, we can choose:

$$\varepsilon \leq \mathcal{O}\left(\frac{1}{q^{3/2}d^{1/2}L_1^{3/2}}\right).$$

As a result, the convergence rate is $\mathcal{O}(\sqrt{\frac{d}{T}})$. The proof is completed.

$\square$