# OpenReview forum: "Enhancing Zeroth-Order Fine-Tuning for LLMs via Gradient-Guided Subspace Selection"
_ICLR.cc/2026/Conference — Submitted to ICLR 2026_

### Official Review · Reviewer_8tvk · 2025-10-27

**Soundness:** 2
**Presentation:** 2
**Contribution:** 2
**Rating:** 2
**Confidence:** 3

**Summary:**

This paper proposes SVD-0, a zero-order (ZO) optimization method for LLM finetuning that aims to achieve more accurate gradient estimates while retaining the variance reduction benefits of subspace methods. Specifically, SVD-0 applies singular value decomposition (SVD) to the gradient estimates derived from an existing ZO optimizer, thereby constructing principled subspace projection matrices. The superiority of SVD-0 over various ZO methods is demonstrated by extensive experiments on a variety of language modeling tasks across model sizes and architectures.

**Strengths:**

* The paper is well-written, with a comprehensive literature review and clearly articulated motivations for the proposed algorithmic design.
* The proposed method exhibits strong empirical performance across diverse tasks, model sizes, and architectures, supported by extensive experimental evidence. Furthermore, the authors provide a detailed investigation of the effects of algorithmic hyperparameters.

**Weaknesses:**

1. The primary concern lies in the substantial overlap with SubZero (Yu et al., 2024), in both algorithmic design and theoretical analysis. The paper would be significantly strengthened by a clearer delineation of its novel contributions.
    - The overall procedure of SVD-0  is highly similar to SubZero, except for the acquisition of matrices $U$ and $V$ in (2). However, several shared components, such as layer-wise perturbations and periodic updates, are also included in the claimed contributions, which may be an overstatement.
    - As is acknowledged in the paper, the theoretical derivation follows the same approach used in (Yu et al., 2024).

2. There are issues in the statement and proof of Theorem 1. The theorem statement does not clearly distinguish the target optimization error $\epsilon$ from the perturbation scale $\varepsilon$. As a convention, the target accuracy should be an arbitrary tolerance, rather than the designated perturbation scale. Moreover, in the proof of Theorem 1 in Appendix C, a factor $1/T$ is missing before the summation at line 1126, which affects the final complexity expression.
3. The empirical justification for the claimed advantages of the SVD-derived subspace could be more direct. To substantiate the claim of improved gradient accuracy, it would be beneficial to provide a direct comparison of the cosine similarity between the full gradient estimates and the true first-order gradients, rather than solely comparing their singular value vectors. Additionally, a quantitative analysis of the gradient estimate variance is needed to empirically substantiate the claim that the method achieves better variance reduction than random subspace approaches.
4. Since Adam is more common than SGD in practical fine tuning, it is recommended to apply the proposed method together with Adam in order to strengthen the practical contribution.
5.  There are several formatting issues.
    - Tables 6, 7, 8, and 9 are placed too close to the main text.
    - There is an incorrect figure reference on line 125. It should be Figure 1 instead of Figure 3.
    - There is a typo in (3). The term $f(\theta)_t$ should be $\nabla f(\theta_t)$.


Reference
Ziming Yu, Pan Zhou, Sike Wang, Jia Li, and Hua Huang. Subzero: Random subspace zeroth-order optimization for memory-efficient llm fine-tuning. arXiv preprint arXiv:2410.08989, 2024.

**Questions:**

1. Could the authors include dominant parameter-efficient fine-tuning baselines, such as LoRA and its advanced variants, to provide a clearer comparison against other widely-used memory-efficient adaptation methods? Additionally, have the authors explored integrating the proposed method with LoRA?
2. Could the authors provide training loss curves plotted against wall-clock time to compare the empirical convergence of SVD-0 against SubZero? While the paper establishes matching theoretical convergence rates in terms of iteration steps, SVD-0 incurs a higher per-step computational cost due to the SVD operation. This wall-clock time comparison is essential to determine which method converges faster in actual training time.
3. How sensitive is the SVD-0 framework to the choice of the underlying ZO gradient estimator? Testing SVD-0 with other ZO estimators would clarify if this is a general subspace strategy or a heuristic tightly coupled to MeZO's specific estimation properties.

---

> ### Author Response · Authors · 2025-11-28
>
> **AW1: Differentiation from subzero**
>
> Although SVD-0 and SubZero exhibit structural overlap, SVD-0 is explicitly constructed to exploit our observation that SVD can uncover relationships between ZO and FO gradients. Both approaches formally employ the matrices $U$ and $V$ to construct perturbations, yet the underlying sources differ fundamentally. SVD-0 adopts this framework because SVD factors a matrix into $U S V^t$, allowing us to generate perturbations that are directly guided by gradient information. SubZero, in contrast, uses a similar form to produce low-dimensional random perturbations. Consequently, even though the mathematical forms resemble each other, the motivations and starting assumptions are entirely distinct. Our main contribution is the incorporation of gradient information into ZO methods for the first time, which yields significant gains in fine-tuning performance. Other components—such as layer-wise perturbations (to limit training overhead) and periodic updates (to further cut training costs and improve fine-tuning)—are also critical to the effectiveness of SVD-0.
>
> **AW2: Issues in theorem 1 statement and proof**
>
> In the earlier version, we incorrectly wrote $\varepsilon$ on the right-hand side where $\epsilon$ should have appeared. Throughout the rest of the derivation, as well as in the proof given in the appendix, we consistently maintained a clear distinction between these two quantities. Consequently, the resulting conclusions remain valid.
>
> Furthermore, we note that the factor $\frac{1}{T}$ was omitted in the proof of Theorem 1 in Appendix C. The final complexity analysis is based on this corrected formula. Thank you for pointing out.
>
> **AW3: Suggested empirical analyses**
>
> We measured the cosine similarity between ZO and FO gradients. For comparability, the reported cosine similarity values are averaged across the full fine-tuning trajectory and rescaled to lie within the [0, 1] interval.
>
> |Method|Cosine Similarity|
> |:-|:-|
> |MeZO|0|
> |SubZero|0.73|
> |SVD-0|1|
>
> This direct comparison of cosine similarity shows that SVD-0 provides a more accurate approximation of the FO gradient.
>
> We also carried out experiments to measure the variance of the gradient estimates. The outcomes are summarized below.
>
> |Method|Variance|
> |:-|:-|
> |MeZO|5.5|
> |SubZero|5.6|
> |SVD-0|5.0|
>
> The results show that SVD-0 achieves the lowest gradient estimation variance, further demonstrating its superiority.
>
> **AW4: Performance on Adam optimizer**
>
> We conducted additional experiments using the Adam optimizer.
>
> |Method|SST-2|WIC|RTE|
> |:-|:-|:-|:-|
> |MeZO(Adam)|75.6|55.8|54.2|
> |SVD-0(Adam)|92.9|62.4|75.5|
>
> Our findings indicate that, when combined with the Adam optimizer, SVD-0 consistently outperforms MeZO. This implies that SVD-0 is particularly well-suited to real-world fine-tuning settings that use Adam.
>
> **AW5: Formatting issues**
>
> Thank you for pointing out these issues. We will adjust the positioning of Tables 6, 7, 8, and 9, revise the figure citation on line 125, and correct the typo in Equation (3).
>
> **AQ1: Performance integrating with LoRA**
>
> Because vanilla LoRA is fine-tuned with Adam, we ran experiments comparing SVD-0 with SGD and vanilla LoRA with Adam.
>
> |Method|Accuracy(\%)|Memory(GB)|
> |:-|:-|:-|
> |LoRA (Adam)|93.2|12.926|
> |SVD-0 (SGD)|93.0|4.891|
>
> SVD-0 (FT) and LoRA (Adam) attain similar performance on the SST-2 dataset; however, SVD-0 (FT) requires substantially less memory than LoRA (Adam). This highlights the main benefit of SVD-0 as a zeroth-order approach.
>
> Additionally, we combined SVD-0 with LoRA and conducted experiments on OPT‑13B.
>
> |Method|SST-2|RTE|CB|BoolQ|WSC|WIC|MultiRC|
> |:-|:-|:-|:-|:-|:-|:-|:-|
> |MeZO (LoRA)|92.2|74.4|69.6|75.2|64.4|59.7|58.2|
> |SVD-0 (LoRA)|93.8|76.9|71.4|76.0|65.4|61.4|59.3|
>
> As shown by the results, SVD-0 (LoRA) consistently outperforms MeZO (LoRA) across all datasets, suggesting that integrating SVD-0 with LoRA yields a substantial performance boost.
>
> **AQ2: Loss curves plotted against wall-clock time**
>
> Because of OpenReview’s limitations, we cannot include images directly.  The relevant image is referenced in Appendix C, Figure 3 of the revised paper.
> We observe that while SVD-0 requires slightly more time than SubZero, it achieves lower training loss in both the initial and later phases. This suggests that, when considering total training time, SVD-0 actually converges more quickly.
>
> **AQ3: Test with other ZO estimators**
>
> In this setting, we substitute MeZO with alternative ZO estimators to supply the gradient information needed by SVD‑0. In particular, we adopt SPSA and NES (Natural Evolution Strategies) as the gradient estimators.
>
> |Estimators|Accuracy(\%)|
> |:-|:-|
> |MeZO|92.2|
> |SPSA|90.5|
> |NES|90.3|
>
> We observe that while SPSA and NES perform slightly worse than MeZO, they still achieve strong performance. This indicates that SVD‑0 is a broadly applicable subspace method that does not rely on MeZO specifically.

---

> ### Comment · Reviewer_8tvk · 2025-11-28
>
> Thank you to the authors for the explanations and experiments, which address most of my concerns. I will consider raising my score. Below is my comment on the rebuttal:
> 1. In AQ1, could you please explain why compare LoRA and SVD-0 with different optimizer setting (SGD vs Adam). As I understand it, using Adam will inevitably consume more memory (about 3 x) than an SGD-type update, which may obscure the improvements attributed to SVD-0.

---

> > ### Author Response · Authors · 2025-11-29
> >
> > In AQ1, we used Adam as the optimizer for LoRA and SGD for SVD-0 because these are the standard choices for each method. To ensure that SVD-0’s memory efficiency wasn’t simply due to the use of a different optimizer, we also tested LoRA with SGD. The results indicate that LoRA’s memory usage with SGD remains significantly higher than SVD-0 with SGD, and only slightly less than when LoRA uses Adam. This demonstrates that the optimizer is not the main factor influencing LoRA’s memory consumption, likely because LoRA has so few trainable parameters that Adam’s overhead is minimal. These findings reinforce that SVD-0’s memory efficiency is intrinsic to the method itself, rather than an artifact of optimizer selection.
> >
> >
> > | Method    |  Accuracy(%)  | Memory(GB) |
> > | :---------|  :------- | :------- |
> > | LoRA (Adam)  |  93.2  | 12.926   |
> > | LoRA (SGD)  |  93.2  | 11.068   |
> > | SVD-0 (SGD)     |  93.0   | 4.891  |

---

### Official Review · Reviewer_4QZ1 · 2025-10-28

**Soundness:** 2
**Presentation:** 3
**Contribution:** 2
**Rating:** 2
**Confidence:** 4

**Summary:**

This paper presents SVD-0 as a memory-efficient method for fine-tuning large language models using zeroth-order (ZO) optimization. It improves gradient estimation by periodically applying singular value decomposition (SVD) to ZO gradients. The authors claim that this creates accurate layer-wise projection matrices to capture the most important optimization directions, thereby reducing the high variance commonly seen in earlier methods such as MeZO, SubZero, and LOZO. As a result, SVD-0 delivers outstanding performance on benchmarks like SuperGLUE tasks, applicable to models including OPT (1.3B/13B), RoBERTa-large, and Qwen-1.8B. It maintains low memory usage and provides proven convergence.

**Strengths:**

The paper is well-motivated. Furthermore, the narrative is remarkably clear and easy to follow, which makes the technical content highly accessible.

**Weaknesses:**

1) While the authors present a well-motivated goal of leveraging gradient information to guide parameter updates more effectively, their method relies entirely on historical zeroth-order gradients derived from MeZO. It is important to note that zeroth-order gradients diverge significantly from true gradients, and accurate gradient estimation under a single SPSA perturbation remains highly challenging[1]. As a result, although the intention is sound, the technical execution does not appear to constitute a substantial breakthrough.

2) the claimed contribution seems incremental when compared to SubZero[2]. The core distinction lies in the construction of the low-rank subspace: whereas SubZero employs random projection for singular value decomposition, SVD-0 utilizes historical zeroth-order gradients from MeZO. However, since these historical gradients essentially form a scalar-projected random matrix, performing SVD on them would only affect the singular values—not the orthogonal matrices. This implies that SVD-0 and SubZero are functionally equivalent as stochastic zeroth-order optimizers in subspace projection.

3) This work can be viewed as a minor variation of SubZero from a theoretical perspective, where the novel theoretical contributions remain unclear.

4) In ICLR submission，use \citet when the author is the subject of the sentence. Use \citep for all other citations, see line331-342.

[1] Malladi, Sadhika, et al. "Fine-tuning language models with just forward passes." Advances in Neural Information Processing Systems 36 (2023): 53038-53075.

[2] Yu, Ziming, et al. "Subzero: Random subspace zeroth-order optimization for memory-efficient llm fine-tuning." (2024).

**Questions:**

see the Weaknesses.

---

> ### Author Response · Authors · 2025-11-28
>
> **AW1: Does not constitute the substantial breakthrough claimed**
>
> Although zero-order gradients deviate considerably from the exact gradients, our preliminary study shows that, after SVD, their similarity becomes fairly high, indicating the presence of common, exploitable components. Moreover, our experiments demonstrate that the SVD-0 approach substantially improves fine-tuning performance.
>
>
>
> **AW2: Functional equivalence with subzero**
>
> The two approaches are not functionally equivalent. In SubZero, the $U$ and $V$ matrices come from applying QR decomposition to two random matrices, not from an SVD. These matrices are independent of each other and serve only to produce a fully random low-rank perturbation. In contrast, our approach derives the $U$ and $V$ matrices by performing SVD on gradient information. Specifically, the SVD-0 method uses a single-step gradient matrix (rather than a history of gradients), obtained from a complicated forward pass that encodes substantial structural information. This setup violates the assumptions of standard random matrix theory. Consequently, the orthogonal matrices produced by SVD on this gradient matrix differ markedly from those obtained from a purely random matrix, as they capture directions that are directly tied to the underlying gradient structure.
>
>
>
>
> **AW3: Novel theoretical contributions remain unclear**
>
> The two approaches are fundamentally different. In SubZero, the $U$ and $V$ matrices are obtained by performing QR decomposition on two independent random matrices, rather than using SVD. Their only role is to introduce a random, low-rank perturbation. By contrast, our method constructs $U$ and $V$ directly from gradient information using SVD. In particular, the SVD-0 variant runs SVD on a single-step gradient matrix (not on an accumulated history of gradients). This gradient matrix encodes rich structural information induced by the forward pass and does not fall within the standard setting of purely random matrix theory. As a result, the orthogonal matrices produced by SVD on this gradient matrix reflect gradient-specific structure, whereas those derived from random matrices are structureless and purely random.
>
>
>
>
> **AW4: Citation format**
>
>
> Thank you for highlighting this. We will use `\citet` when referring to authors as the subject of a sentence, and `\citep` for all other citation contexts.

---

### Official Review · Reviewer_NhqK · 2025-10-31

**Soundness:** 3
**Presentation:** 3
**Contribution:** 2
**Rating:** 4
**Confidence:** 3

**Summary:**

This paper introduces SVD-0, a zeroth-order (ZO) optimization method for improving large language model (LLM) fine-tuning by exploiting the low-rank structure of gradients. Motivated by the observation that ZO gradients share similar spectral properties with first-order (FO) gradients, the authors propose to extract a low-rank subspace via SVD on the approximated ZO gradients and restrict perturbations to this subspace. This approach can help reduce the approximation variance of ZO gradient estimation, thereby achieving more accurate gradient approximation and improved fine-tuning performance. Extensive experiments on a range of LLMs and benchmarks are provided to demonstrate the effectiveness of SVD-0 compared to prior ZO methods such as MeZO, SubZero, and LOZO.

**Strengths:**

The paper presents a well-motivated extension of GaLore-style low-rank subspace projection (Zhao et al., 2024) to the zeroth-order (ZO) optimization framework. By incorporating low-dimensional structure of gradients without backpropagation, it effectively improves the efficiency of ZO gradient estimation compared to prior methods that use random subspace directions. The experimental results are well designed.

**Weaknesses:**

While the preliminary study suggests that ZO gradients share similar spectral properties with exact FO gradients, the evidence given is somewhat limited.  Additional analysis would make the argument more convincing. The algorithm description is also unconvincing, particularly regarding the use of different perturbation matrices (see Questions below). The performance improvement over prior methods appears relatively limited.

**Questions:**

- In the prestudy of Section 3, could the authors provide more details about the experiments, such as the chosen rank and how the cosine similarity is computed (i.e., what the x-axis in Figure 1 represents and whether the cosine similarity is averaged across all singular vectors)? The result shows a cosine similarity around 0.45, but is this high enough to claim that the estimated ZO and FO gradients are similar? How does the similarity change for higher-order singular vectors and with more batches used for ZO estimation?

- In line 19 of Algorithm 3, the weights are updated with a new perturbation matrix, which is inconsistent with the SPSA in MeZO formulation where the same perturbation direction should be used for both gradient estimation and parameter update. Intuitively it doesn't really make sense to measure the difference of function values along one direction but apply the update in another direction. Could the authors clarify on this?

- The authors mention that the additional time required for SVD operations is negligible even though the time complexity of SVD is $O(n^3)$. Could the authors clarify on this?

- In Eq (3), should $U_t^\top f(\theta)_t V_t$ be $U_t^\top \nabla f(\theta_t)V_t$? Also, some symbols like $\theta$, $U$, and $V$ are written in non-bold fonts which is inconsistent with the rest of the paper.

---

> ### Author Response · Authors · 2025-11-28
>
> **AW1: Limited evidence on gradient similarity**
>
> In Appendix B, we present experiments involving several dimensionality reduction techniques. These results indicate that SVD is most effective at extracting components shared across the two gradients, thereby further underscoring the similarity between ZO and exact FO gradients.
> We also analyzed the cosine similarity between gradients generated by different ZO methods and the FO gradient. During fine-tuning, we periodically collected the true gradients for a selected subset of parameters, along with the gradients estimated by the ZO method. We then computed the cosine similarity between the ZO and FO gradients. The reported cosine similarity values are averaged over the full fine-tuning run and, for clearer comparison, scaled to lie within the [0, 1] interval.
>
> | Method    |  Cosine Similarity  |
> | :---------|  :-------|
> | MeZO      |  0    |
> | SubZero   |  0.73       |
> | SVD-0     |  1       |
>
> The findings show that the SVD-based approach attains the greatest similarity. This implies that isolating the common components between ZO and FO gradients through SVD yields a more accurate approximation of the FO gradient, thereby providing additional support for the perspective put forward in our preliminary study.
>
> **AW2: Unconvincing algorithm description**
>
> For a detailed explanation, please refer to AQ2.
>
> **AW3: Limited performance gain**
>
> Sorry for any confusion. While the performance gains achieved by our method may appear modest, as reported in Table 2, they are clearly larger than those obtained by other ZO variants. We recognize that our approach does not consistently outperform all other ZO techniques across all datasets. Yet this limitation is shared by the alternative ZO methods, which likewise fail to deliver top performance across all datasets and tasks. For example, although S-MeZO achieves the highest accuracy on 4 of the 7 classification datasets, its overall classification results remain inferior to those of SVD-0. Moreover, SVD-0 substantially outperforms S-MeZO on both multiple-choice and generation tasks. As summarized in Table 2, our SVD-0 method achieves the strongest performance on most datasets and task types, highlighting its overall advantage and robust generalization.
>
> **AQ1: Pre-study details**
>
> In this experiment, we fix the rank at 24 when truncating the singular value vectors. In Figure 1, the x-axis corresponds to the number of fine-tuning training steps, while the y-axis shows the cosine similarity at each step. This cosine similarity is computed over a subset of parameters rather than being averaged across all parameters.
>
> In terms of cosine similarity obtained with SVD (see Appendix B), this method produces values substantially higher than those obtained with other decomposition approaches. We therefore identify SVD as the most effective method for extracting the common components shared by ZO and FO gradients. As for fine-tuning performance, SVD-0 delivers the strongest results across most datasets and tasks, further reinforcing this conclusion.
>
> We further examined how similarity changes under different ranks and batch sizes. In particular, we used ranks of 8, 24, and 48, and batch sizes of 1, 2, 4, 8, and 16. The corresponding results are shown below.
>
> | Rank\\Batch Size | 1 | 2 | 4 | 8 | 16 |
> | :----|  :----- | :--- | :-- | :-- | :-- |
> | 8| 0.87| 0.86| 0.50| 0.72| 0.67|
> |24| 0.85| 0.84| 0.45| 0.70| 0.63|
> |48| 0.85| 0.84| 0.45| 0.70| 0.63|
>
> The results indicate that as the rank decreases, the similarity drops slightly but remains largely stable. Adjusting the batch size causes more pronounced variations in cosine similarity, without a clear upward or downward trend. Overall, the similarity stays consistently high.
>
> **AQ2: Inconsistency in perturbation matrix usage**
>
> Although we technically resample at every step, we fix a random seed $s^t$ and reuse it for the resampling. Consequently, the resulting perturbation matrix is identical to that from the previous step, thereby keeping the measurement and update directions consistent. This procedure follows the same strategy as MeZO, as described in Section 2.1 of the MeZO paper.
>
> **AQ3: Explanation for negligible SVD overhead**
>
> Although SVD has a time complexity of $O(n^3)$, this is on par with the $O(n^3)$ complexity of the matrix multiplications that dominate forward propagation in our models. In addition, our periodic update scheme invokes SVD only a limited number of times throughout the entire fine-tuning procedure, further limiting any additional computational burden. Taking these aspects into account, we consider the SVD-related overhead to be negligible, a claim corroborated by our experimental observations.
>
> **AQ4: Notation errors in Eq (3)**
>
> Thank you for pointing out this issue. In Equation (3), the expression should indeed be written as $U_t^\top \nabla f(\theta_t)V_t$. We also fix the inconsistent font styles used for symbols such as $\theta$, $U$, and $V$.

---

### Official Review · Reviewer_3KtC · 2025-11-01

**Soundness:** 3
**Presentation:** 3
**Contribution:** 3
**Rating:** 6
**Confidence:** 3

**Summary:**

The paper proposes SVD-0, a zeroth-order (ZO) fine-tuning method that replaces random low-dimensional subspaces with gradient-guided ones computed via singular value decomposition (SVD) of ZO gradient estimates. Prior ZO subspace (low-rank) methods constrain perturbations to random low-rank subspaces, but their arbitrary projection matrices can misalign with the gradients’ intrinsic low-rank structure; meanwhile conventional ZO like MeZO suffers from high variance in billion-parameter spaces. SVD-0 periodically (every $F$ steps) computes layer-wise projection matrices $(U,V)$ from the estimated gradients, then perturbs parameters with low-rank updates $\tilde Z = U Z V^\top$ to improve gradient estimation accuracy while preserving ZO’s memory efficiency with minimal overhead.

**Strengths:**

1. The pre-study clearly demonstrates that zeroth-order (ZO) gradient estimates preserve the low-rank structure of the true gradients. This provides empirical motivation for constructing projection matrices from these estimates rather than from random perturbations.
2. The method design is well organized: (1) SVD-based acquisition of gradient-guided projection matrices $(U, V)$ from estimated gradients, and (2) generation of low-rank perturbations $\tilde{Z} = U Z V^T$ within these subspaces. Algorithm 1–3 explicitly describe this process.
3. Experimental results show SVD-0 outperforming MeZO, SubZero, and LOZO across multiple models and benchmarks.
4. The method maintains the low-memory footprint of zeroth-order optimization.
5. Provides theoretical analysis of convergence.

**Weaknesses:**

1. Standard deviations are missing for OPT experiments, even though they are provided for RoBERTa-large.
2. Line 125 references Figure 3, when I think it’s meant to refer to Figure 1.
3. The computational efficiency of SVD-0 can become a bottleneck for larger models due to the high time complexity of the SVD operation. While Table 1 should specify the model and experimental settings used for comparing the computational costs of different methods, it is also important to report these results on larger models (with 7B+ parameters).

**Questions:**

Please refer to the weaknesses.

---

> ### Author Response · Authors · 2025-11-28
>
> **AW1: Missing standard deviation in OPT experiments**
>
> Fine-tuning OPT models requires substantially more computational resources than fine-tuning RoBERTa-large. Consequently, we did not run multiple OPT instances with different random seeds and therefore do not report standard deviation values. However, to show that SVD-0 consistently delivers superior performance across various random seeds on OPT, we performed the following experiment, in which we fine-tuned the OPT-1.3B model on the SST-2 dataset.
>
> | Method\\Seed  | 0   |   42     | 123	     |
> | :---------| :------- | :------- | :------- |
> | MeZO      | 90.9     | 88       | 89.2     |
> | SVD-0     | **93.0**   | **90.1** | **90.4** |
>
> These results show that SVD-0 consistently outperforms MeZO across all random seeds. We have added the results to Table 18 in Appendix C of the paper.
>
> **AW2: Incorrect figure reference**
>
> Sorry for the mistake. The correct reference should be to Figure 1. We will update this and carefully review the document for any other similar mistakes.
>
>
>
> **AW3: Clarification on setup for cost comparison and larger model results**
>
> In the experiments summarized in Table 1, each model was fine-tuned on its respective task for 20,000 steps. In particular, the WIC and ReCoRD datasets were paired with the OPT‑1.3B model, whereas the FiQA‑SA and TFNS datasets were paired with the Qwen‑1.8B model. Since running 70B-parameter models is highly resource-intensive, we instead report runtime results from 13B-parameter models. These runtime experiments were carried out on the MultiRC dataset.
>
> | Method    |  Time   |
> | :---------|  :------- |
> | MeZO      |  1125.3   |
> | SubZero   |  1149.5   |
> | SVD-0     |  1154.8   |
>
> We evaluated the computational cost (in minutes) for OPT-13B, as illustrated above, and found that the processing time increased by less than 3\%. The runtime is therefore comparable to that of the smaller models. This indicates that our approach does not substantially increase the computational overhead when applied to larger models.
> We have added the results to Table 1 in the paper.

---

### Official Review · Reviewer_bjHR · 2025-11-03

**Soundness:** 2
**Presentation:** 2
**Contribution:** 2
**Rating:** 6
**Confidence:** 2

**Summary:**

LLMs are large networks, and even first-order gradient descent can be too computationally and memory-intensive to finetune. Zeroth-order (ZO) skips backprop by probing the model with tiny nudges and seeing the loss change. This saves memory, but the “which way to move” signal is very noisy in huge models.

The paper's key idea is that, instead of searching in random directions, they perform an SVD to find a few strong directions, then explore mostly within that tiny set. Specifically, they 1) get a ZO gradient estimate using a standard method 2) run SVD on that estimate to get two small matrices $U, V$. They define a low-rank subspace. 3) Conduct updates with low-rank perturbations $U Z V^{\top}$ instead of full random noise. Then they recompute $U, V$ every F steps to balance compute and adaptability.

This works because ZO and true gradients share structure, so SVD on the ZO estimate reveals helpful directions. We can then search mostly where progress actually happens.

The paper shows that, in tests on SuperGLUE with OPT-13B and 1.3B, SVD-0 is consistently among the top ZO methods and often best overall versus MeZO and others.

SVD is $O\left(n^3\right)$ and it can be expensive. But they show that the overhead stays low, costing like 7% longer only. This is possible becauses they do 1) Layer-wise SVD, 2) Periodic refresh. and 3) No per-step extra work.

**Strengths:**

I was thinking that the observed ZO $\iff$ FO spectral alignment was interesting; it makes the SVD-guided subspace a reasonable lever. I believe their method's value is low integration cost and stable top-2 placement.

The cost of the proposed method is not significant, indicating it is practical.

**Weaknesses:**

The biggest weakness of this paper is the modest empirical gains it achieves relative to state-of-the-art baselines. Some margins are really narrow. The improvements are consistent but at best modest on core benchmarks.

One small caveat is that projection quality depends on ZO gradient precision, so that smaller models can yield noisier $U$ and $V$. But I believe zeroth-order gradients are mostly for tuning large networks.

**Questions:**

1. I wonder how the cost scales from 1.3B to 13B to 70B parameters.

2. I wonder how this method would work under quite noisy ZO gradients.

3. How can you pick $r$ and $F$ automatically from live signals?

4. How often does the subspace become stale on domain shifts?

---

> ### Author Response · Authors · 2025-11-28
>
> **AW1: Limited performance improvement**
>
> Sorry for the confusion. While the performance gains of our method reported in Table 2 may appear modest, they are still substantially larger than those obtained by other ZO variants. We recognize that our method does not consistently outperform all other ZO techniques across all datasets. However, this variability is not specific to our approach—no existing ZO method achieves the best performance across all datasets and tasks. For example, S-MeZO achieves the top results on 4 of 7 classification datasets, yet its overall classification performance is still worse than SVD-0. Moreover, SVD-0 significantly surpasses S-MeZO on multiple-choice and generation tasks. In summary, as shown in Table 2, our SVD-0 method achieves state-of-the-art results across most datasets and task categories, underscoring its overall advantage and strong generalization capability.
>
>
>
>
>
>
>
> **AW2: Gradient noise in small models**
>
> We add Gaussian noise to the ZO gradients to mimic inaccuracies in the gradient estimation. The experimental results show that the SVD-0 method preserves robust performance even when the ZO gradients are noisy. For comprehensive experimental results, please refer to AQ2.
>
>
>
> **AQ1: Computational cost on larger model**
>
> Because running 70B models demands substantial computational resources, we report time-cost experiments using 13B models instead. These experiments were conducted on the MultiRC dataset to demonstrate that our method does not significantly increase computational overhead.
>
> | Method    |  Time     |
> | :---------|  :------- |
> | MeZO      |  1125.3   |
> | SubZero   |  1149.5   |
> | SVD-0     |  1154.8   |
>
> We measured the computational time (in minutes) for OPT-13B as presented above. We found that the processing time rose by less than 3\%. This runtime overhead is comparable to that observed for smaller models.
>
>
>
>
> **AQ2: Performance under noisy zero-order gradients**
>
> We injected Gaussian noise with different variances into the zero-order gradients and ran the following experiments. Specifically, we added zero-mean Gaussian noise to the ZO gradients, setting its variance to be $K$ times that of the projected gradients to mimic noisy environments. We then fine-tuned the OPT‑1.3B model on the SST-2, RTE, and WIC datasets. The corresponding results are summarized below.
>
>
>
> | K    |  SST-2  | RTE  | WIC |
> | :----|  :----- | :--- | :-- |
> |No noise  |  92.2   | 68.2 | 56.3|
> | 0.5  |  87.5   | 68.6 | 56.6|
> | 1    |  88.9   | 66.1 | 56.4|
> | 2    |  87.5   | 66.4 | 55.0 |
>
> Overall, as noise variance increases, the model’s performance declines slightly, demonstrating that the SVD-0 method remains robust under noisy conditions. In addition, when no noise is present or the noise level is very low, the model attains its highest performance, indicating that SVD-0 successfully exploits the informative components of the ZO gradients to improve fine-tuning.
>
>
>
> **AQ3: Automatic Selection of $r$ and $F$**
>
> As evidenced by the experimental results in Tables 4 and 5, our approach reliably preserves strong performance except when $r$ or $F$ are chosen to be excessively extreme. Overall, changes in $r$ and $F$ have minimal impact on performance, suggesting that it is not strictly necessary to automatically tune these parameters.
>
> **AQ4: Subspace Accuracy Duration during Domain Shift**
>
> As shown in the analysis of $F$ in Table 4, training performance only deteriorates noticeably when $F$ is set to an extremely large value, leading to rare subspace updates. This indicates that the subspace stays accurate for a relatively long time.

---

### Author Response · Authors · 2025-12-01

We sincerely appreciate the thoughtful feedback and constructive conversations from the reviewers. Below, we provide a brief overview of our latest clarifications, updates, and commitments:

- Additional experiments:
    - Robustness to Noisy ZO Gradients: SVD‑0 exhibits substantial resilience, as adding zero-mean Gaussian noise to the ZO gradients leads to only minor performance degradation, even as the noise level increases. The best results are obtained when the noise level is very low or zero. (Reviewer bjHR)

    - Stable Results Across Random Seeds: SVD‑0 consistently outperforms alternatives across various random seeds, highlighting its robustness. (Reviewer 3KtC)

    - Gradient Similarity and Variance: Among all decomposition methods, SVD achieves the highest cosine similarity between ZO and FO gradients, making it the most faithful surrogate to FO. Moreover, SVD-0 attains the smallest variance in gradient estimates, underscoring its superior accuracy. (Reviewer NhqK & 8tvk)

    - Adam Optimizer: SVD‑0 (Adam) consistently outperforms MeZO (Adam), underscoring its suitability for real‑world Adam-based fine-tuning. (Reviewer 8tvk)

    - Other ZO Estimators: Replacing MeZO with either SPSA or NES still yields robust accuracy, indicating that SVD‑0 functions as a flexible subspace approach that does not depend on the specific ZO estimator used. (Reviewer 8tvk)

    - Integration with LoRA: Across multiple OPT‑13B datasets, SVD‑0 (LoRA) reliably outperforms MeZO (LoRA). At the same time, SVD‑0 attains accuracy on par with standard LoRA while requiring substantially less memory (4.891 GB versus 12.926 GB), highlighting its superior memory efficiency. (Reviewer 8tvk)


- Efficiency:
    - Runtime: SVD‑0 incurs less than a 3\% increase in execution time over MeZO/SubZero on OPT‑13B, representing a minimal overhead considering the improved accuracy. (Reviewer bjHR & 3KtC)

    - Wall-clock Convergence: Loss-versus-time analyses reveal that SVD‑0 reaches lower loss than SubZero during both early and later training phases, despite a marginally higher per-step cost. This reflects its faster real-time convergence. (Reviewer 8tvk)


- Clarifications:
    - Novelty and “Breakthrough” Concern: Unlike SubZero’s reliance on random QR, SVD‑0 leverages SVD on single-step gradient matrices to derive orthogonal bases that capture meaningful gradient structure beyond just random matrix assumptions. The observed empirical improvements further demonstrate the significance of this contribution. (Reviewer 4QZ1 & 8tvk)

- Contribution: SVD-0 advances ZO fine-tuning by utilizing gradient-informed subspaces, consistently delivering strong performance across a range of tasks and models with little additional cost.

We sincerely appreciate the reviewers’ time, consideration, and assessment. We reiterate that SVD-0 offers a practical and theoretically well-founded approach for memory-efficient fine-tuning of large language models in ZO settings.

---

### Meta-Review · Area_Chair_pGzx · 2026-01-07

**Summary:**

During the rebuttal, the reviewers acknowledged the authors' effort, specifically regarding secondary concerns. However, several key concerns remain insufficiently resolved.

**Novelty and Conceptual Distinction:** Multiple reviewers (most notably Reviewer 4QZ1 and Reviewer NhqK) questioned the novelty and conceptual distinction between the proposed method and prior work, specifically SubZero (Yu et al., 2024). The reviewers felt that applying SVD to gradient-informed matrices rather than random projections was not proven to be a fundamentally new optimization behavior, but rather an incremental variant.

**Modest Performance Gains:** Although the empirical analysis was expanded, reviewers remained concerned that the performance gains were relatively modest compared to the added complexity of the SVD step. Reviewer NhqK noted that the improvement over prior methods appeared limited, making the trade-off questionable.

**Theoretical Analysis:** Reviewer 4QZ1 raised significant issues with the theoretical formulation, particularly regarding Theorem 1 and the distinction between target optimization error and perturbation scale. Despite the rebuttal, these theoretical gaps were not fully closed to the reviewers' satisfaction.

**Reviewer Concerns:**

The authors addressed concerns about the stability of the method, added runtime and memory comparisons on larger models, and clarified notational issues (e.g., typos in Equation 3) and algorithm descriptions. However, the differentiation from SubZero remains the primary sticking point, as the authors did not convince reviewers that this constituted a distinct algorithmic contribution. Also, the issues pointed out by Reviewer 4QZ1 regarding the proof structure were not definitively resolved in the discussion, and the performance gains remain a valid concern.

**Reviewer Scores:**

Reviewer 4QZ1 would likely have maintained the most critical score, and Reviewer 8tvk indicated the willingness to raise the score (likely from 2 to 4). Other reviewers would likely have maintained their scores.

---

### Decision · Program_Chairs · 2026-01-26

Reject